# Structural and functional analysis of the minimal orthomyxovirus-like polymerase of Tilapia Lake Virus from the highly diverged *Amnoonviridae* family

Benoit Arragain [1], Martin Pelosse[1], Albert Thompson [1,2] & Stephen Cusack [1] ✉

Tilapia Lake Virus (TiLV), a recently discovered pathogen of tilapia fish, belongs to the *Amnoonviridae* family from the *Articulavirales* order. Its ten genome segments have characteristic conserved ends and encode proteins with no known homologues, apart from the segment 1, which encodes an orthomyxo-like RNA-dependent-RNA polymerase core subunit. Here we show that segments 1–3 encode respectively the PB1, PB2 and PA-like subunits of an active heterotrimeric polymerase that maintains all domains found in the distantly related influenza polymerase, despite an unprecedented overall size reduction of 40%. Multiple high-resolution cryo-EM structures of TiLV polymerase in pre-initiation, initiation and active elongation states, show how it binds the vRNA and cRNA promoters and performs RNA synthesis, with both transcriptase and replicase configurations being characterised. However, the highly truncated endonuclease-like domain appears inactive and the putative cap-binding domain is autoinhibited, emphasising that many functional aspects of TiLV polymerase remain to be elucidated.

Tilapia Lake Virus (TiLV) was first isolated from moribund fish following mass die offs of Nile Tilapia (*Oreochromis niloticus*) in fish farms in Israel[1]. Subsequent analysis showed that it is an enveloped, negative-sense single-stranded RNA virus with ten genome segments, each with at least one open-reading frame (ORF)[2]. Segment 1 encodes a protein of 519 residues containing motifs characteristic of the orthomyxovirus RNA-dependent RNA polymerase (RdRp) polymerase basic protein 1 (PB1) subunit, whereas the other nine ORFs exhibit no sequence homology to any other known protein. This, together with the presence of conserved, quasi-complimentary 5′ and 3′ termini on each segment led to the tentative classification of TiLV as a novel orthomyxo-like virus[2]. TiLV outbreaks have been reported in at least 16 countries located on four continents. The severity of TiLV outbreaks varies from asymptomatic with no associated mortality to over 90%

mortality[3–5]. TiLV therefore poses a significant threat to the global tilapia aquaculture industry, which provides proteinaceous food to millions of people, especially in the developing world.

TiLV is now defined as a new species, *Tilapia tilapinevirus*, in the Tilapinevirus genus, within the *Amnoonviridae* family, which together with the *Orthomyxoviridae*, form the *Articulavirales* order[6]. Whilst the phylogeny of different isolates[7], detection and pathogenesis of the virus and possible treatments by vaccination[8] or drugs[9] are now being extensively studied, the molecular virology of TiLV is poorly understood, partly due to the current lack of reverse genetics or mini-genome systems[10]. In addition, the uniqueness of the putative TiLV encoded proteins, apart from a recognisable polymerase PB1-like subunit, precludes hypotheses as to protein function based on sequence homology or structure prediction using Alphafold[11] or

[1]European Molecular Biology Laboratory, 71 Avenue des Martyrs, CS 90181, 38042, Grenoble Cedex 9, France. [2]Present address: The Francis Crick Institute, London, UK. ✉e-mail: cusack@embl.fr

RoseTTAfold[12]. However, using a bioinformatics approach based on comparative features of orthomyxoviral proteins, segment 2 and 3 proteins were predicted to be additional polymerase subunits[13].

Here, we set out to structurally characterize the TiLV encoded replication machinery, which should comprise at least an RdRp, potentially heterotrimeric and a nucleoprotein. We used co-expression of all TiLV ORFs in insect cells to identify proteins co-purifying with the segment 1-encoded protein, the putative core RdRp subunit. This enabled purification of a heterotrimeric complex comprising proteins from segments 1–3, which could be structurally characterised by high-resolution cryogenic electron microscopy (cryo-EM) (Table 1). We show that this heterotrimer resembles the influenza virus polymerase (FluPol), with segments 2 and 3 being respectively polymerase basic protein 2 (PB2)- and polymerase acidic protein (PA)-like in domain structure and fold, even though the TiLV polymerase (155 kDa) is remarkably only 61% of the overall molecular FluPol mass (255 kDa). Recombinant TiLV polymerase specifically binds the conserved 5′ and 3′ genome and anti-genome ends, with mode A and mode B promoter structures similar to other segmented negative strand RNA viruses[14–18]. Furthermore, it is active in RNA synthesis with an appropriate primer. Our structural and functional characterisation of TiLV polymerase is a first step towards a detailed understanding of the transcription/replication machinery of this intriguing virus.

## Results

### Identification of the Tilapia Lake Virus polymerase

TiLV (*Amnoonviridae* family) segment 1 ORF contains conserved orthomyxo-like RdRp motifs suggesting that it is a PB1-like subunit[2]. To identify other viral proteins that can interact with the segment 1-encoded protein we added a purification tag at its C-terminus and co-expressed it in insect cells together with the nine other putative TiLV proteins using the EMBacY bacmid (see Methods and Supplementary Fig. 1). After an initial affinity purification step, segment 1-encoded protein (57 kDa) co-eluted with two other proteins, matching the molecular weights of segments 2 (51 kDa) and 3 (47 kDa), suggesting formation of a heterotrimeric complex. However, this initial complex was unstable upon further purification, likely due to suboptimal positioning of the purification tag (see Methods and Supplementary Fig. 1). A new expression construct encoding segment 1, 2 and 3 proteins, with segment 2 harbouring a TEV-cleavable deca-histidine purification tag on its C-terminus, enabled pure and stable heterotrimeric complex, suitable for further structural and functional studies, to be obtained (Fig. 1a).

### Reconstitution of TiLV polymerase with the promoter and cryo-EM structure determination

Viral polymerases from segmented negative stranded RNA viruses (sNSVs) typically interact directly with the conserved and quasi-complementary 5′ and 3′ viral genome ends, usually referred to as the promoter[15,17,18]. We therefore sought to identify TiLV promoter regions to be able to reconstitute an active TiLV heterotrimeric complex in vitro. Unfortunately, the previously highlighted viral promoter regions for each segment[2] do not correspond to those deduced from the deposited negative sense RNA sequences (e.g., Genbank sequences for the ten genome segments of Tilapia lake virus isolate Til-4-2011, KU751814.1 to KU751823.1). The correct vRNA and complementary (cRNA) extremities for each segment show several features distinct from classical orthomyxoviral promoter sequences. The TiLV 5′ and 3′ vRNA ends are quasi-complementary over 15 nucleotides and, like bunyavirus promoters, are of the same length, without the bulge seen in the panhandle representation of the influenza promoter (Fig. 1b). Exceptionally, there is no complementarity between the first nucleotides of the 5′ and 3′ ends. We next synthesized 40-mer vRNA and cRNA loops, each corresponding to the joining of the first and last 20 nucleotides of the 5′ and 3′ vRNA/cRNA ends of segment 9 (Fig. 1b).

Using single particle cryo-EM, we solved multiple sub-3Å resolution structures of reconstituted TiLV polymerase-promoter complexes that allowed unambiguous model building of the complete polymerase and promoter structures, in different functional states (Table 1, Supplementary Table 1, Supplementary Notes 1–5).

### Overall structure of TiLV polymerase and comparison with influenza polymerase

TiLV polymerase is composed of PB1-, PB2- and PA-like subunits, encoded by segments 1–3, respectively. The TiLV heterotrimer is only 60% of the molecular weight of the distantly related orthomyxovirus FluPol (155 kDa versus 255 kDa). Remarkably, each subunit maintains the same domains as found in FluPol subunits, but is minimised in size, except the RNA-synthesis performing RdRp core (Fig. 1c). The unusual presence of three zinc ions denoted Zn1, Zn2 and Zn3 (identified by their tetrameric co-ordination with cysteine or histidine residues) might be one way by which domains are stabilized within the overall smaller architecture of TiLV polymerase. Zn1 is located in the PA C-terminal domain (PA-C), while Zn2 and Zn3 are located in the cap-binding domain-like (CBD-like) of PB2 (Fig. 1d). Furthermore, TiLV polymerase can take up the two functionally distinct overall configurations that are analogous to the FluPol transcriptase[19,20] and replicase[21–24] conformations (Fig. 1d, e). Due to the structural relatedness of the TiLV and influenza polymerases, despite high sequence divergence, we henceforth drop the '-like' qualification in future reference to TiLV polymerase subunits and domains, except for endonuclease-like (ENDO-like) domain and CBD-like domain, whose functionality remains to be clarified (see below).

### TiLV PA subunit

Comparison of the TiLV and FluPol PA subunits reveals that despite sharing no sequence homology and being 42% smaller (417 residues versus 713 residues for A/H17N10 FluPol), the structural fold and organization are conserved (Fig. 2a). Like FluPol, TiLV PA subunit is divided into three domains: a minimal N-terminal ENDO-like domain (1-100, 100 residues, 195 in A/H17N10); a short linker domain (101-135, 35 residues, 61 in A/H17N10); and the PA-C domain (136-417, 282 residues, 457 in A/H17N10) (Fig. 2a). TiLV ENDO-like domain is a highly truncated version of FluPol endonuclease and does not display any residues that could be involved in nuclease activity (Fig. 2b). Indeed, FluPol residues H41, E80, D108, E119 that coordinate divalent metal ions and K134, which performs nucleophilic attack, correspond to F5, Y29, D43, E56 and R62, respectively, in the TiLV ENDO-like domain (Fig. 2c). In vitro endonuclease activity assays revealed that TiLV polymerase, in the presence of various divalent metal ions is unable to cleave single stranded RNA, whether or not bound to the 5′ and 3′ vRNA ends, in contrast to FluPol (Fig. 2d). The inactivity of TiLV ENDO-like domain is reminiscent of Thogoto and Dhori virus polymerases, whose ENDO-like domains also have non-functional active sites[25], raising the questions as to if and how TiLV performs cap-dependent transcription and the functional role of the degenerate domain (Supplementary Fig. 2).

### TiLV PB1 subunit

TiLV PB1 subunit (57 kDa, 515 residues) encompasses the expected RdRp sub-domains, fingers (1-182, 224-265), palm (183-223, 266-337), thumb (338-454) and bridge (455-515). They are structurally analogous to the FluPol PB1 domains, but are overall 44% smaller (Fig. 3a). TiLV PB1 structural minimization examples are numerous. The FluPol PB1 α-helices (84-123, 40 residues) are replaced by extended chain in TiLV PB1 (46-73, 28 residues) (Fig. 3b), the FluPol PB1 β-ribbon (162-222, 61 residues) is completely absent in TiLV PB1 (107-137, 31 residues) (Fig. 3c). More strikingly, the FluPol PB1 priming loop, which is an essential structural element in the orthomyxovirus genome replication initiation mechanism[19,26], is completely absent in TiLV PB1 (Fig. 3d).

**Table 1 | Summary of TiLV polymerase structures**

| Structure No. | Short name | Resolution | RNA/promoter mode | Description | PDB/EMDB |
|---|---|---|---|---|---|
| **vRNA initiation** | | | | | |
| 1 | Full vRNA initiation with CTP | 2.73 Å | vRNA/A | Full mode A structure with 3' end in the active site and incoming CTP at +1 | PDB 8PSN EMD-17857 |
| 2 | Core vRNA initiation with CTP | 2.40 Å | vRNA/A | Core only mode A structure with 3' end in the active site and incoming CTP at +1 | PDB 8PSO EMD-17858 |
| **cRNA pre-initiation** | | | | | |
| 3 | Core cRNA pre-initiation mode A | 2.65 Å | cRNA/A | Core only mode A structure with 3' end in the active site | PDB 8PSQ EMD-17860 |
| 4 | Core with endo cRNA pre-initiation mode A | 2.80 Å | cRNA/A | Core with endo, mode A structure with 3' end in the active site | PDB 8PT7 EMD-17869 |
| **cRNA mode B** | | | | | |
| 5 | Closed core with endo cRNA mode B | 2.83 Å | cRNA/B | Core with endo, mode B structure with 3' end in the secondary site (weak duplex) | PDB 8PSS EMD-17861 |
| **vRNA pre-initiation** | | | | | |
| 6 | Core vRNA pre-initiation mode A | 3.18 Å | vRNA/A | Core, mode A structure with 3' end in the active site | PDB 8PSU EMD-17862 |
| **vRNA elongation** | | | | | |
| 7 | Full vRNA elongation with CpNHpp | 2.96 Å | vRNA | Full elongation structure stalled with CpNHpp at +1 | PDB 8PSX EMD-17864 |
| 8 | Full vRNA elongation with CpNHpp and additional mode B promoter | 2.42 Å | vRNA/B | Full elongation structure stalled with CpNHpp at +1 with additional mode B promoter | PDB 8PSZ EMD-17865 |
| **vRNA mode B** | | | | | |
| 9 | Full with open core vRNA mode B | 2.59 Å | vRNA/B | Full mode B structure with 3' end in the secondary site (strong duplex) | PDB 8PT2 EMD-17866 |
| 10 | Open core with rotated endo-NLS vRNA mode B | 2.73 Å | vRNA/B | Core with rotated Endo-NLS mode B structure (open, strong duplex) | PDB 8PTH EMD-17871 |
| 11 | Closed core with rotated endo-NLS vRNA mode B | 2.86 Å | vRNA/B | Core with rotated Endo-NLS mode B structure (closed, weak duplex) | PDB 8PTJ EMD-17872 |
| **Replicase** | | | | | |
| 12 | Replicase initiation with CpNHpp | 2.90 Å | vRNA/B | Core with rotated Endo-NLS and CBD (replicase conformation) mode A with 3' end in the active site and incoming CpNHpp at +1 | PDB 8PT6 EMD-17868 |
| **Pre-termination** | | | | | |
| 13 | Full vRNA pre-termination with GpCpp | 3.13 Å | vRNA | Full elongation structure stalled with GpCpp at +1 close to the template 5' hook | PDB 8QZ8 EMD-18772 |

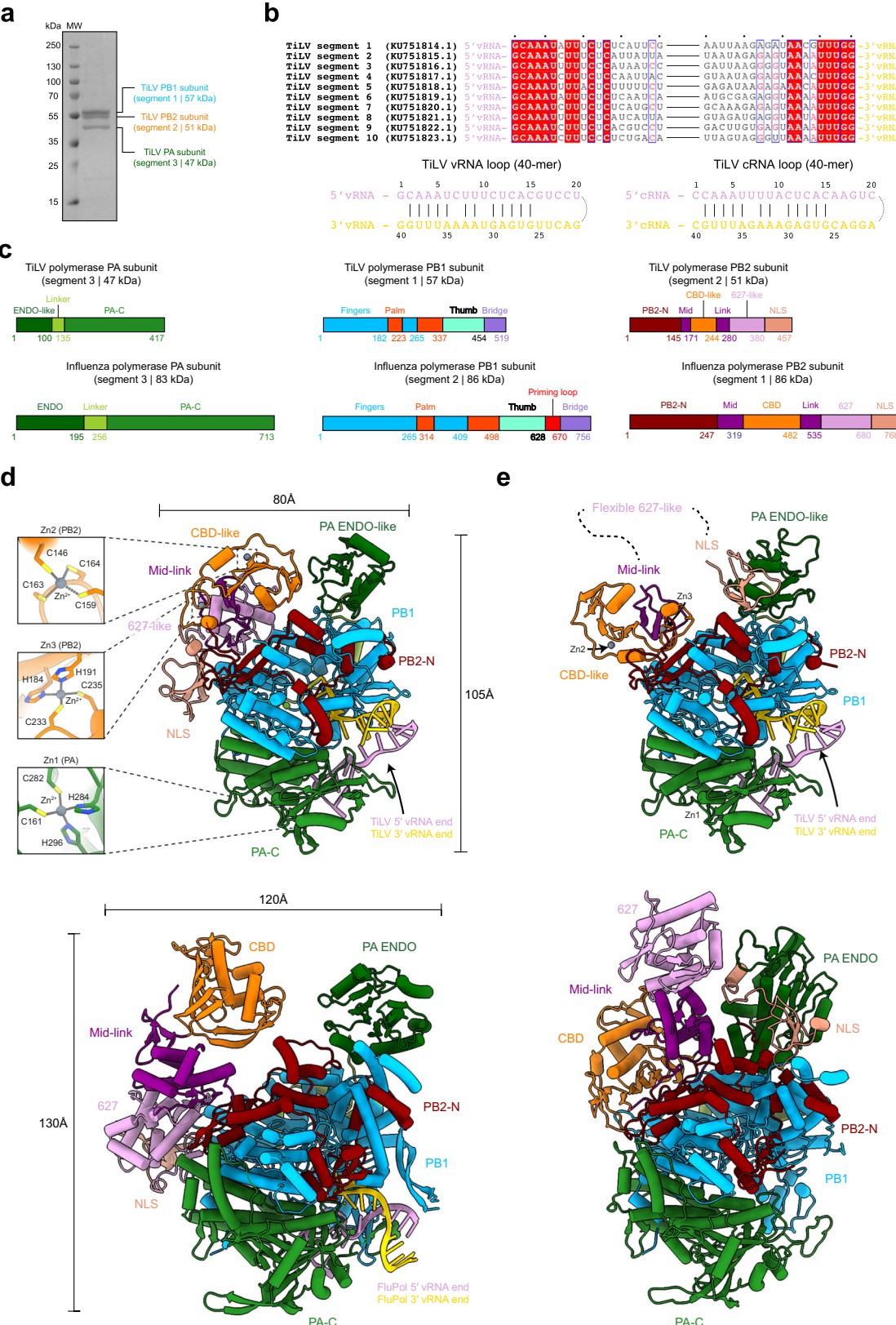

Despite being minimal, the TiLV PB1 subunit contains all the conserved RdRp functional motifs (Fig. 3e), as previously noted[2]. Motifs A (208-223) and C (285-295) are implicated in divalent metal ion coordination through catalytic aspartic acids D213, D289 and D290. Motif B (263-278) is characterized by an *Articulavirales*-conserved methionine rich loop (262-GGMLMMFN). It is implicated in stabilization of the incoming NTP at the +1 position (M266), in translocation and in discrimination of NTPs from dNTPs (N270). Motif D (311-323) contributes to a positively charged entry tunnel for NTPs via lysine residues K318 and K319, in concert with K216 (motif A) and K151 and R155 (motif F). Motif E (326-335) stabilizes motif C through hydrophobic interactions (L327, I329).

**Fig. 1 | TiLV polymerase purification, promoter characterization and structural comparison with influenza polymerase. a** SDS-PAGE analysis of purified TiLV polymerase heterotrimeric complex. The molecular ladder (MW) is shown on the left side of the gel. TiLV polymerase subunits are indicated on the right side of the gel with PB1 (segment 1, 57 kDa), PB2 (segment 2, 51 kDa) and PA (segment 3, 47 kDa) subunits, respectively, coloured in blue, orange and green. Source data are provided as a Source Data file ($n = 6$ independent experiments). **b** Sequence alignment of the 20 first 5′ and 3′ end nucleotides from each TiLV genome segment. Corresponding GenBank numbers are indicated for the ten genome segments of TiLV isolate Til-4-2011. Derived vRNA and cRNA loops (40-mer) from the segment 9 are shown with 5′ ends in pink and 3′ ends in gold. **c** Schematic comparison of the domain structures of TiLV polymerase and FluPol subunits. Rectangle sizes are proportional for each corresponding polymerase domain. **d** Cartoon representation of TiLV polymerase and FluPol A/H17N10 (PDB:4WSB) in the transcriptase conformation with respective domains coloured as in **c**. The three TiLV zinc-binding sites (Zn1, Zn2 and Zn3) and their coordinating residues are displayed on the left. 5′ and 3′ vRNA ends are coloured as in (**b**). TiLV polymerase and FluPol are aligned with each other using PB1 core as reference. **e** Cartoon representation of TiLV polymerase and FluPol A/H5N1 (PDB:6QPF) in the replicase conformation, coloured as in (**c**) and (**d**). TiLV PB2 627-like domain is flexible. TiLV polymerase and FluPol are aligned using PB1 core as reference.

Motif F (138-176) is also implicated in template stabilization (I157, R165).

## TiLV PB2 subunit

TiLV PB2 subunit (457 residues) is 40 % smaller than A/H17N10 PB2 (760 residues), but similarly divided into the PB2 N-terminal domain (PB2-N, 1-145) and an array of PB2-C domains. These include the split mid-link (mid: 146-171, link: 245-280), CBD-like (172-244), 627-like (281-380) and NLS (381-457) domains (Fig. 4a). TiLV PB2-N contains a minimal two α-helical lid domain (77-117, 41 residues) compared to the larger FluPol one (80 residues), which is implicated in template-product duplex separation during elongation[27] (Fig. 4b).

The minimized CBD-like domain is inserted into the mid-link domain and, unusually, both domains contain putative cap-binding residues. F253 and W263 from the TiLV link domain form a potential aromatic sandwich comparable to PB2/H357 and F404 in FluPol, between which stacks the m⁷G cap, and TiLV D251 is equivalent to FluPol PB2/E361, which ensures guanine specificity (Fig. 4c). However, in TiLV, R217 is sandwiched between F253 and W263 and makes a salt bridge with D251. The auto-inhibition of potential cap-binding by R217 in TiLV is reminiscent of Thogoto virus (THOV) polymerase[25] and severe fever with thrombocytopenia syndrome virus (SFTSV, also called Dabie bandavirus) L protein[28,29]. For THOV CBD-like domain, PB2/R344 is sandwiched between an aromatic and an arginine and for SFTSV-L CBD, R1843 is between a pair of aromatic residues (Supplementary Fig. 3). THOV CBD-like domain appears to be non-functional[25], but in the case of SFTSV, it has been shown structurally that R1843 can swing out of the way to allow cap-binding during transcription initiation (Supplementary Fig. 3). To test whether TiLV polymerase can bind cap, we used immobilized m⁷GTP beads. We found that vRNA promoter bound TiLV polymerase is not able to bind the beads, in conditions where FluPol can (Fig. 4d). It remains to determine whether cap binding is possible under different conditions.

The TiLV 627-like domain (281-380, 99 residues) is substantially smaller than the equivalent FluPol domain (144 residues) but has the same topology (Fig. 4e), whereas the TiLV NLS domain (381-457, 76 residues) is of comparable size to the FluPol NLS domain (79 residues) (Fig. 4f). The folded part of the TiLV NLS domain is immediately followed by a putative NLS motif 427-KKRK, similar to FluPol 740-KRKR. In all TiLV polymerase cryo-EM maps, PB2 C-terminal residues 425-457 remain unseen, presumably due to their flexibility.

In FluPol, a stable PB1-PB2 inter-subunit interaction is formed by a helical bundle involving the C-terminus of PB1 and the N-terminus of PB2[30], reinforced in the transcriptase conformation by a helix from the endonuclease domain[19]. In TiLV, similar interactions are observed although much reduced in scale (not shown). Perhaps to compensate, in TiLV there is a unique extra C-terminal extension to PB1 (494-515) that wraps around and reinforces the small PB2 helical lid.

## TiLV polymerase binding to the vRNA and cRNA promoters

As for other sNSVs polymerases[14–18], TiLV polymerase can bind both the vRNA and cRNA promoter in two distinct modes (Fig. 5). In mode A, the 3′ end is directed into the polymerase active site for initiation of RNA synthesis (Fig. 5a; Supplementary Fig. 4), whereas in mode B, the 3′ end is docked on the surface of the TiLV polymerase core, in the so-called "secondary site" (Fig. 5e; Supplementary Fig. 5).

In both modes, the vRNA 5′ end nts 3-AAAUCUU-9 form a compact stem-loop structure (hook) that binds in a pocket covered by the PA arch (202-212) (Fig. 5b). A3 and A4 respectively base pair with U9 and U8, while A5 to C7 form a loop with U6 stacking on A4 (Fig. 5b; Supplementary Figs. 4a, 5a). The two first 5′ end nucleotides have no density and are presumably flexible. The 5′ hook interacts with the polymerase through backbone interactions and base stacking (e.g., PA/H207 on C7 and PA/Y202 under the A3-U9 base pair). Only splayed out A5 and U6 are base-specifically recognized by PA/L355 and S357 and PA/S206 and H207, respectively (Fig. 5b; Supplementary Fig. 4a). The 5′ hook position and shape is similar to that observed in other sNSV viruses but is unusually small, comprising only 7 nts compared with 10 for FluPol, LACV-L/SFTSV-L, LASV-L and up to 12 for HTNV-L (Supplementary Fig. 6).

In the vRNA mode A pre-initiation state, a strong 5-mer distal duplex is formed by 5′ end nucleotides C11 to C15 base-pairing with 3′ end nts G11 to G15 (Fig. 5c). U10 connects the hook to the distal duplex, with a sharp bend being induced by the binding of K263 to the phosphates of both U10 and C11 (Fig. 5b, c, Supplementary Fig. 4a). The 3′ end enters the active site in a highly structured way (Fig. 5c, d, Supplementary Fig. 4a) with A9 and U5 being splayed base-specific recognition pockets. A8 stacks on A7 and U4 and U3 are stacked on each other with U3 being base-specifically recognized. Finally, G1 and G2 are stacked on each other at the −1 and +1 positions respectively in the polymerase active site (Fig. 5d, Supplementary Fig. 4a).

There are six differences in the vRNA and cRNA promoter sequence for segment 9, which however do not significantly affect the overall mode A or mode B promoter structure (Supplementary Figs. 4a, 5a). Notably, all base-specifically recognized nucleotides (5′ A5 and U6 and 3′ U3, U5 and A9 as well as G2 in mode B, see below) are absolutely conserved in the vRNA and cRNAs of all segments. Only the substitutions from 5′ U10 and 3′ A6 (vRNA) to 5′ A10 and 3′ G7 (vRNA) cause a locally different nucleotide packing against the distal duplex in mode A, other base substitutions being accommodated without perturbation. The trajectory of the 3′ end into the active site is identical for vRNA and cRNA, except that for cRNA, C1 and G2 are at the −1 and +1 positions respectively, rather than G1 and G2 for vRNA. This suggests that for both cRNA and vRNA, replication is initiated terminally. Whereas other sNSVs polymerases have a 'buffer zone' that allows the template to extend and then retract[31], while maintaining the distal duplex during prime-and-realign, the extensive and tight binding of the 3′ end suggests this might not be the case in TiLV.

In the mode B pre-initiation state, the 5′ hook binds as in mode A, but 3′ end nts 1-GGUUUA in vRNA are sequestered in the secondary site at the interface of PA-C, PB1 and PB2-N on the surface of TiLV core (Fig. 5e; Supplementary Fig. 5). G1 is stabilized by PA/Q175 and D245, while G2 is base-specifically bound in a pocket, stacking between PA/M229, R233 and PB2/R39, with PB2/D35 and K36 contacting N2 and N7 respectively and the carbonyl of PB1/R394 contacting N2. U3 is sandwiched between PB2/F42 on one side and U4 on the opposite side, with

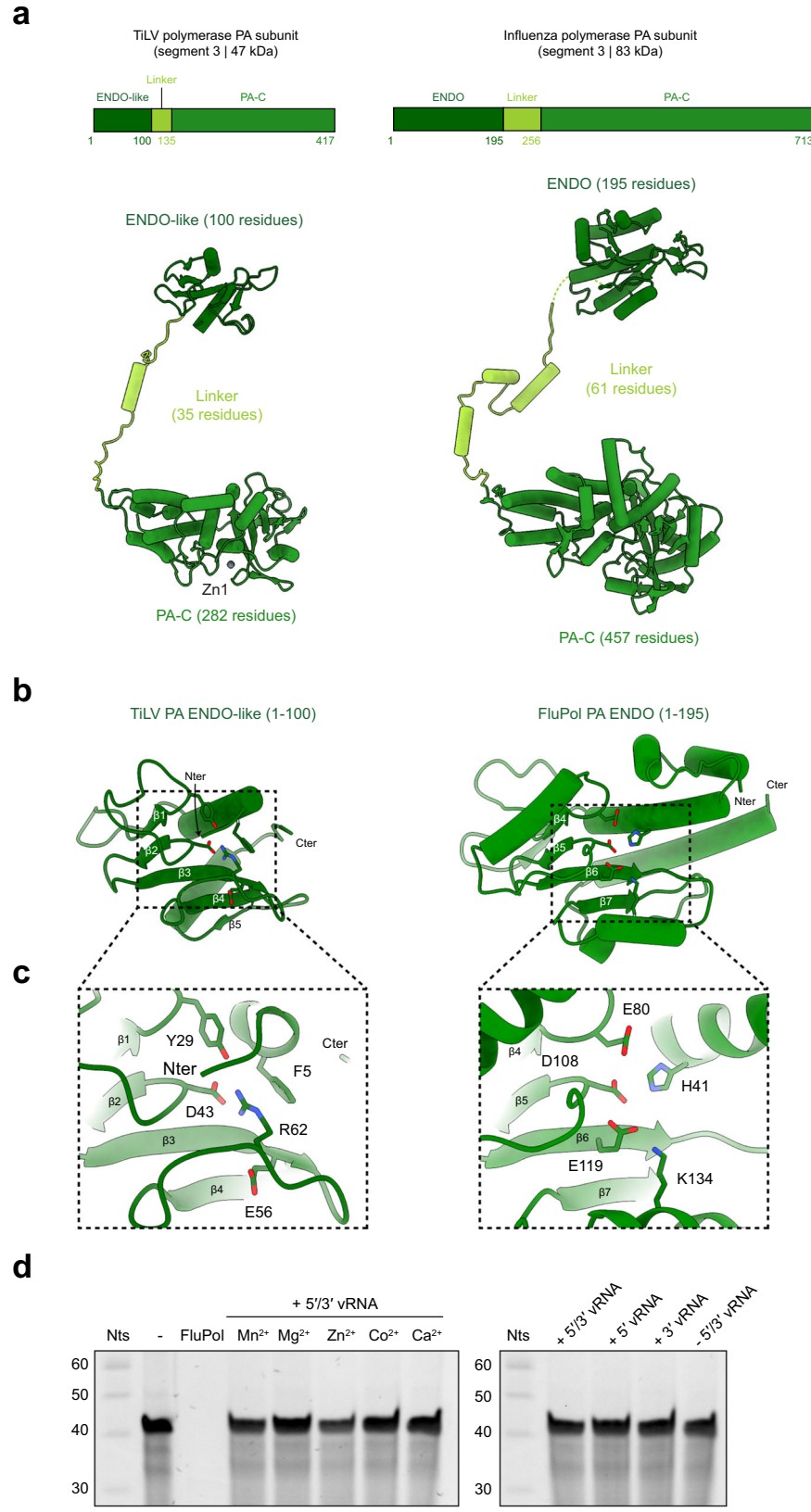

PA/Q237 bridging both G2 and U4 phosphates (Fig. 5f; Supplementary Fig. 5a). Extensive 3D classification allowed two distinct populations of mode B vRNA particles to be isolated (Supplementary Note 4), either with a closed core and ill-defined distal duplex, due to disorder of 3′ end residues A7 to G15 (Supplementary Fig. 5a, b), or with an open core and well-resolved distal duplex (Supplementary Fig. 5a, c). For cRNA,

only the closed core conformation without distal duplex is observed (Supplementary Fig. 5d, Supplementary Note 2). In the open conformation, vRNA 3′ end A6 is stabilized in a pocket formed by PA/T236, T267, T270 and H272 residues. A7 interacts with PA/R265 and stacks with A8. A9 is not visible but its flexibility allows U10 to flip out and stack with PA/R265, the latter being stabilized by R194 (Supplementary

**Fig. 2 | TiLV polymerase PA-like subunit in transcriptase conformation.**
**a** Schematic and cartoon representations of TiLV polymerase and FluPol A/H17N10 (PDB:4WSB) PA subunits extracted from their respective transcriptase conformation. Domains are coloured as in Fig. 1c. **b** Comparison of TiLV PA ENDO-like (1-100) and FluPol A/H17N10 (PDB:4WSB) PA ENDO (1-195) domain structures. The N- (N$_{ter}$) and C-terminus (C$_{ter}$) are indicated. The conserved β-sheets of TiLV polymerase and FluPol were used for domain alignment. **c** Close-up of TiLV ENDO-like and FluPol A/H17N10 (PDB:4WSB) ENDO domain active site. FluPol ENDO catalytic residues are displayed as well as the corresponding residues of TiLV ENDO-like domain. **d** In vitro TiLV PA ENDO-like domain RNA cleavage activity. Left: Urea-PAGE analysis of the effect of different divalent metal ions on TiLV polymerase PA ENDO-like

cleavage activity. Lanes "-" and "FluPol" correspond to negative (no protein) and positive (A/H7N9 FluPol) controls. TiLV polymerase bound to the 5′ (1-15) and 3′ vRNA (1-15) ends, incubated with manganese (Mn$^{2+}$), magnesium (Mg$^{2+}$), zinc (Zn$^{2+}$), cobalt (Co$^{2+}$), or calcium (Ca$^{2+}$), is not able to cleave single-stranded RNA. The decade marker (Nts) is shown on the left of the gel. Right: Urea-PAGE analysis of the effect of vRNA ends on TiLV ENDO-like cleavage activity. TiLV polymerase bound to (i) the 5′ (1-15) and 3′ vRNA (1-15) ends (+5′/3′ vRNA), (ii) the 5′ vRNA end only (+5′ vRNA), (iii) the 3′ vRNA end only (+3′ vRNA) or (iv) no RNA (−5′/3′ vRNA) using manganese (Mn$^{2+}$), is not able to cleave single-stranded RNA. The decade marker (Nts) is shown on the left of the gel. Source data are provided as a Source Data file (*n* = 3 independent experiments).

Fig. 5a, c). The distal duplex is formed, as in mode A, between the 3′ vRNA nucleotides G11 to G15 base-pairing with the 5′ vRNA nucleotides C15 to C11 (Supplementary Fig. 5a, c). It is notable that the mode B duplex is at a significantly different orientation to that observed in mode A (Supplementary Fig. 5e). The two mode B conformations may represent intermediate states in the recycling from mode B to mode A at the end of RNA synthesis to allow a new round of RNA synthesis, as proposed for FluPol[32].

### TiLV polymerase activity: from pre-initiation to initiation

We next investigated whether the polymerase is able to perform de novo replication-like or primed transcription-like activity in vitro. Based on the pre-initiation mode A structures, which shows 3′ terminal nucleotides G1-G2 (vRNA) or C1-G2 (cRNA), respectively, in positions −1/+1 of the active site (Fig. 6a), we hypothesized that de novo replication would likely initiate by the synthesis of the dinucleotide pppCpC or pppGpC, from CTP + CTP or GTP + CTP for vRNA or cRNA, respectively.

To test this hypothesis, we incubated TiLV polymerase with the vRNA loop, CTP, Mg$^{2+}$ and froze the reaction on grids after 1 h. We obtained two cryo-EM structures of TiLV polymerase in the vRNA initiation-like state, one with the complete polymerase (2.73 Å resolution) and one with the polymerase core only (2.4 Å resolution, enabling visualization of many water molecules) (Supplementary Fig. 4d, Supplementary Note 1). The initiation-like state is very similar to the pre-initiation mode A structure, except that a single CTP with two magnesium ions is bound at the +1 active site position, base pairing with template nucleotide G2. Binding of CTP induces only small perturbations in the tilt of the template nucleotides G1 and G2, in the motif C loop, enabling co-ordination of the active site aspartates with the two catalytic magnesium ions and in some motif F residues (PB1/K151 and R155) that co-ordinate the triphosphate (Fig. 6b, Supplementary Fig. 7c). Unlike FluPol[27], there is no major switch in the conformation of the motif B loop backbone, although M266 side chain does flip under the CTP base. The absence of the dinucleotide 5′-pppCpC hybridized to the 3′ vRNA end suggests that it does not form under these conditions or is too weakly bound, bearing in mind that there is no priming loop that might stabilize the pre-catalytic complex. Nevertheless, the template and CTP positions are consistent with TiLV polymerase replication initiation being terminal (Fig. 6a, b; Supplementary Fig. 7).

### Biochemical demonstration of RNA synthesis activity of recombinant TiLV polymerase

We next established in vitro assays to characterize de novo and primed RNA synthesis activity by TiLV polymerase (Fig. 7). Using the 40-mer vRNA loop (v40) and all NTPs, we failed to observe de novo reaction products (Fig. 7a, lane 3). Given that only CTP and not the 5′-pppCpC dinucleotide product was observed in the initiation state structural analysis, this suggests that de novo replication is limited by the initiation step, perhaps due to the absence of a priming loop. We therefore used the dinucleotide CpC as a primer to hybridize to the vRNA template 3′ end (…GG-3′), equivalent to the use of ApG in studies of FluPol RNA synthesis. CpC together with ATP, UTP and Mg2+ only,

resulted in strong production of a product around 10 nts long, likely stalled through lack of CTP to match G11 of the template (Fig. 7a, lane 2). Using CpC with all NTPs yields some stalled product of around 10 nts, and two long products of between 40-50 nts that could correspond to stalling before or read-through of the 5′ hook (Fig. 7a, lane 4).

As we do not know whether cap-binding occurs nor the preferred length of a putative capped primer, to explore transcription-like behaviour we used capped (11-, 12-, 13-mers; all ending with …CC-3′) and uncapped (9-mer, ending with …CC-3′; 10-mer, ending by …CCA-3′) primers to initiate RNA synthesis, with either ATP and UTP only or all NTPs (Fig. 7b). Using both uncapped primers in combination with ATP, UTP, and Mg$^{2+}$, we observe the synthesis of the expected 17-mer product due to stalling at G11, but also slightly longer products that could result from read-through due to misincorporation (Fig. 7b, lane 1-2). With the three capped primers, two main bands are observed that we interpret as the expected capped 19, 20 or 21-mer as well as read-through products two nts longer, likely stalled at G13 (Fig. 7b, lane 3-5). With all NTPs, two long products, differing by about 10 nts are seen with all primers ending in …CC-3′ (Fig. 7b, lane 6, 8-10), again consistent with a possible termination pre- or post-hook. Interestingly, reactions with the uncapped …CCA-3′ primer show lower intensity (with ATP and UTP) or no product (with all NTPs) compared to the uncapped …CC-3′ primer (Fig. 7b, lane 1-2, 6-7). This suggests that U3 of the vRNA template is poorly accessible for hybridization with a primer, consistent with its specific recognition in a pocket distinct from the active site (Fig. 5d). When all NTPs are present, we suggest that CTP binds preferentially at the +1 position, competing out the unfavourable …CCA-3′ primer. These observations highlight the inflexible positioning of the template 3′ end in the active site and the absence of a template buffer zone for TiLV. This is not the case for FluPol, for which capped primers ending by …AG-3′ or …AGC-3′ are equally reactive since the template is advanced by one position into the active site[27,32]. While we do not detect any cap-binding in vitro, reactions with the capped primers show stronger product formation as compared to the 9-mer uncapped primer, which could indicate the optimal hypothetical primer length used by TiLV polymerase for transcription initiation.

To further characterize TiLV polymerase primed RNA synthesis activity, we used a modified vRNA 40-mer template, denoted v40-S, with a C23U mutation (i.e., at position 27 from the 3′ end). With ATP, UTP and CTP, but not GTP, in the reaction mix TiLV polymerase will stall on this template at C19 (Fig. 7c, d). Again, in the absence of any primer, TiLV polymerase is unable to perform de novo replication (Fig. 7c, lane 1). Using CpC, some RNA synthesis occurs but the precise product is unclear (Fig. 7c, lane 2). For the reactions performed using the capped/uncapped primers ending by …CC-3′, synthesis of the expected 28, 30, 31 and 32-mer products occurs with some minor read-through or misincorporation (Fig. 7d, lane 3, 5-7). Consistent with our previous observations, in the presence of CTP, TiLV polymerase is still unable to use a primer ending by …CCA-3′ (Fig. 7d, lane 4). Finally, using all NTPs, a similar activity to that observed with the un-mutated v40 template is observed (Fig. 7d, lane 2-3, 5-7 compared to Fig. 7a, lane 3-4; 8b lane 6, 8-10) with two main products differing by

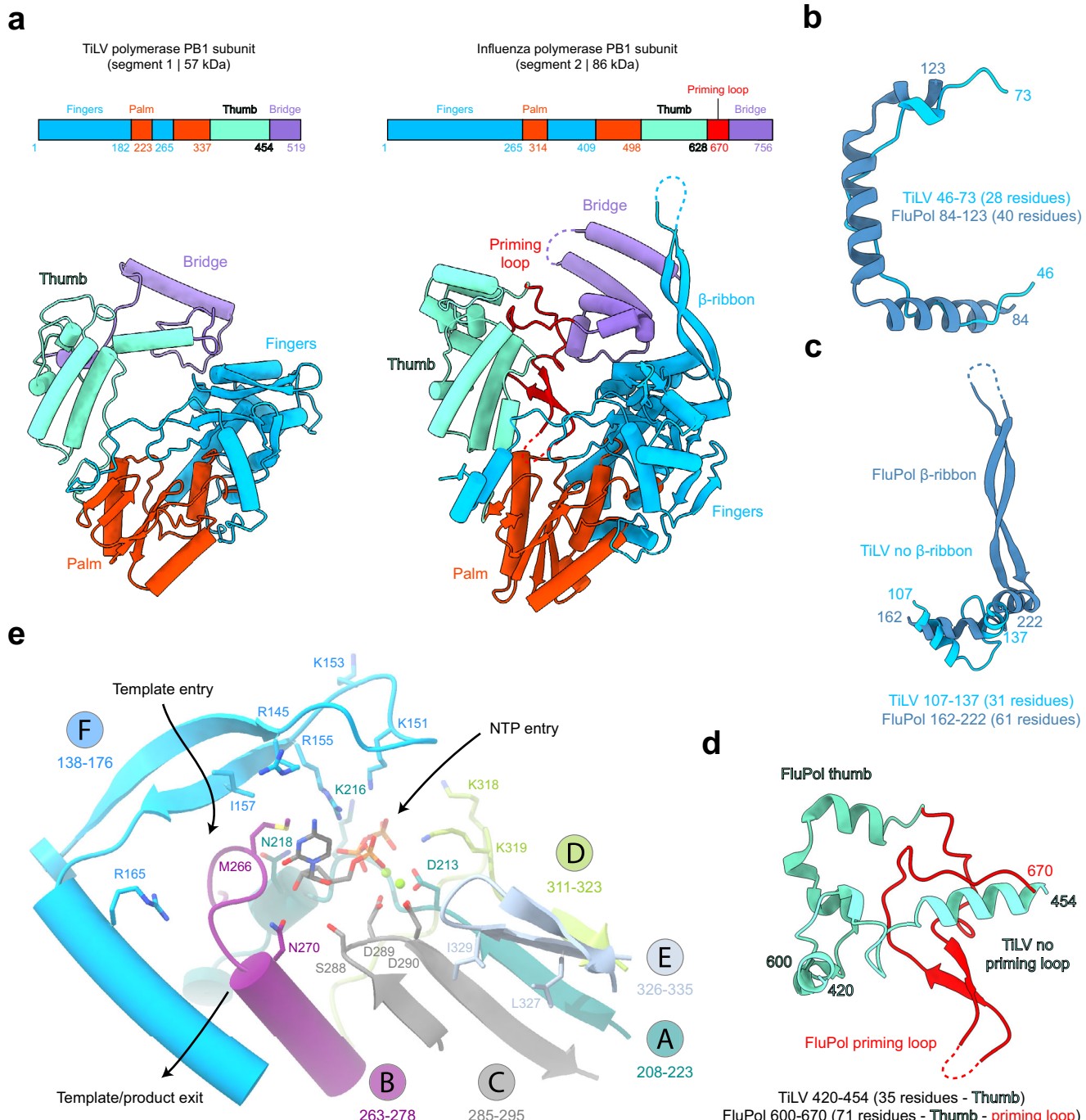

**Fig. 3 | TiLV polymerase PB1 subunit. a** Schematic and cartoon representation of TiLV polymerase and FluPol A/H17N10 (PDB:4WSB) PB1 subunits. Domains are coloured as in Fig. 1c. **b** Miniaturization examples of TiLV polymerase PB1 subunit. Corresponding structures of TiLV PB1 (46-73) and FluPol A/H17N10 (PDB:4WSB) PB1 (84-123) are coloured in light blue and dark blue. **c** Absence of β-ribbon in TiLV PB1 subunit. Corresponding structures of TiLV PB1 (107-137) and FluPol A/H17N10 (PDB:4WSB) PB1 β-ribbon (162-222) are coloured as in (**b**). Flexible residues are shown as dotted line. **d** TiLV PB1 subunit does not have a priming loop. TiLV and

FluPol A/H17N10 (PDB:4WSB) PB1 thumb domains are respectively coloured in light and dark aquamarine. FluPol priming loop is coloured in red. Flexible residues are represented as a dotted line. **e** TiLV PB1 active site. RdRp motifs A, B, C, D, E and F are, respectively, coloured in dark turquoise, magenta, dark grey, light green, light grey and blue. Key residues are displayed and the active site aspartates are circled. In the vRNA initiation state, CTP (dark grey) is in +1 active site position. Magnesium ions (Mg²⁺) are shown as green spheres. The NTP entry, template entry and template/product exit channels are indicated with arrows.

approximately 10 nts that we suggest correspond to termination pre- or post-hook. This interpretation is supported by the fact that only one product is observed when stalling occurs before reaching the hook when only ATP, UTP and CTP are used (Fig. 7c, lane 3, 5-7).

These preliminary results clearly demonstrate that recombinant TiLV polymerase has primed RNA synthesis activity, also

confirmed by the early and late elongation state cryoEM structures described below. However, more detailed analysis is required to identify precisely all the reaction products formed and understand for instance the requirements for efficient de novo initiation in the absence of a priming loop and the mechanism of termination.

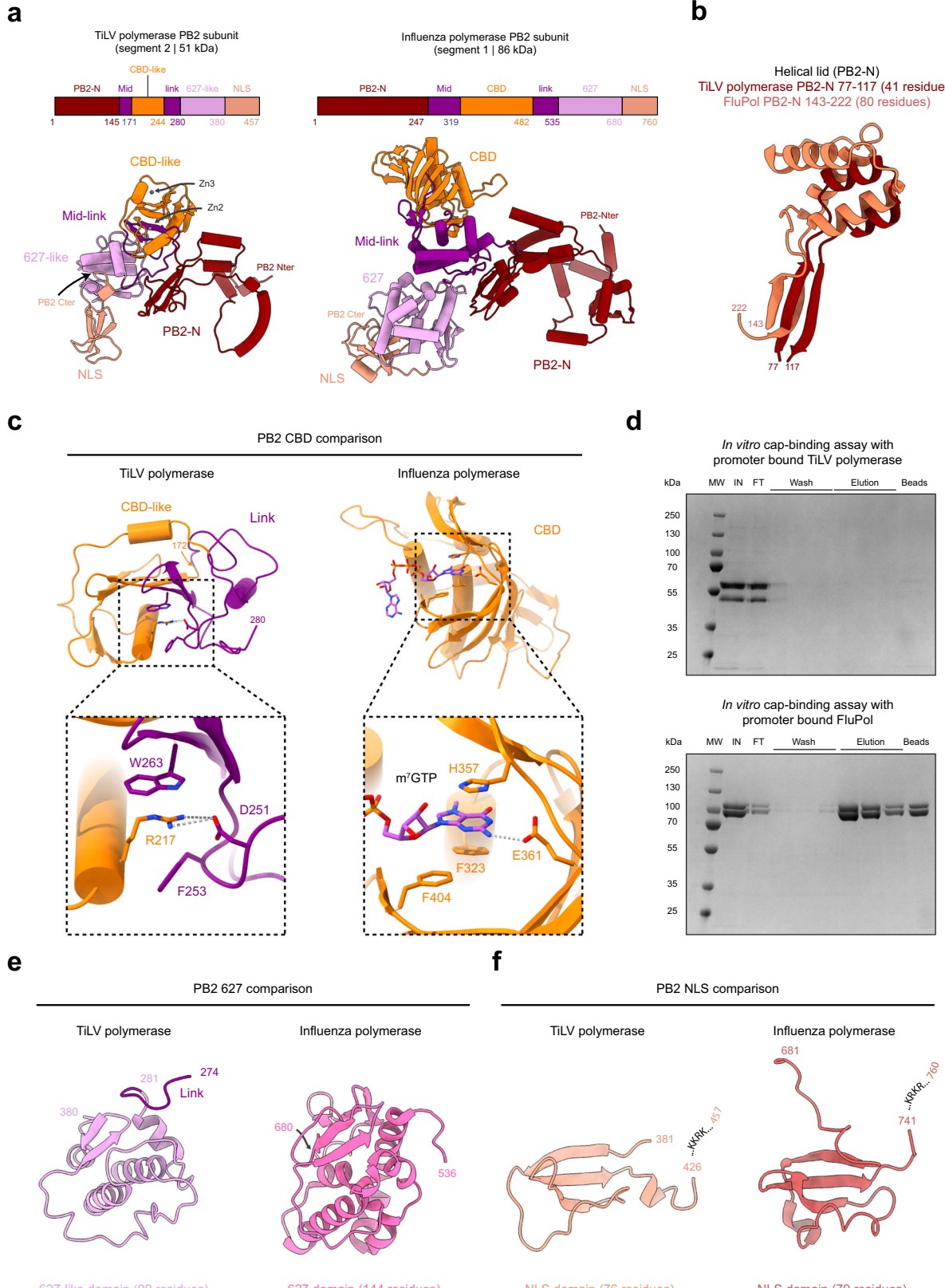

## Structural characterization of TiLV polymerase early and late elongation states

To structurally characterize the early elongation state, TiLV polymerase was incubated with the 40-mer vRNA loop, the capped 13-mer primer ending in …CC-3′, ATP, UTP, Mg$^{2+}$ and CpNHpp. This leads to elongation of the primer by eight nucleotides to become a 21-mer,

before stalling at nucleotide G11 of the template (Fig. 7b, lane 5; Fig. 6c right). After 4 h reaction at 24 C, the sample was applied to EM grids. Two distinct elongation structures of TiLV polymerase were determined from the cryo-EM data at respectively 2.96 and 2.42 Å resolution (Table 1, Supplementary Table 1b, Supplementary Fig. 8a, b, Supplementary Note 4). Both show the full-length TiLV polymerase with

**Fig. 4 | TiLV polymerase PB2 subunit in transcriptase conformation.**
**a** Schematic and cartoon representation of TiLV polymerase and FluPol A/H17N10 (PDB:4WSB) PB2 subunits in transcriptase conformation. Domains are coloured as in Fig. 1c. **b** Miniaturization example of TiLV PB2-N lid. TiLV polymerase lid (77-117), coloured in dark red, is superposed with FluPol A/H17N10 (PDB:4WSB) lid (143-222), coloured in orange. **c** Comparison of TiLV PB2 CBD-like and FluPol A/H7N9 (PDB:7QTL) PB2 CBD. Top: Overall structures of TiLV PB2 CBD-like and link domains (245-280) and FluPol PB2 CBD. Bottom: Close-up views of each respective cap-binding site. TiLV polymerase CBD-like and link domains contain putative cap-binding residues. FluPol CBD is bound to m7GTP (light blue). **d** In vitro cap-binding

assay with promoter bound TiLV polymerase or FluPol A/H7N9. Top: SDS-PAGE analysis of TiLV polymerase cap-binding capacity using immobilized γ-aminophenyl-m7GTP (C10-spacer) beads. The molecular ladder (MW) is shown on the left of the gel. "IN" corresponds to the input. "FT" corresponds to the flow through. Bottom: SDS-PAGE analysis of A/H7N9 FluPol cap-binding capacity using immobilized γ-aminophenyl-m7GTP (C10-spacer) beads annotated as above. Source data are provided as a Source Data file (*n* = 3 independent experiments). **e** Comparison of TiLV PB2 627-like and FluPol PB2 627 domains. **f** Comparison of TiLV PB2 NLS and FluPol PB2 NLS domains. The respective NLS sequence is indicated. In both cases, the NLS C-terminal is not visible due to flexibility.

CpNHpp bound at the +1 position and a product-template duplex filling the active site cavity (Fig. 6c left). RNA sequence identification, unambiguous because of the high resolution of the cryo-EM density, confirms that the observed elongation state is consistent with the biochemical RNA synthesis assay (Fig. 7b, lane 5; Supplementary Fig. 8a–c). The higher resolution structure differs in having an additional 3′ end bound in mode B forming a distal duplex with the 5′ end (see below). Unusually, the product-template duplex comprises 11 base pairs (+1 to −10) (Fig. 6c) where generally for viral polymerases a maximum of 10 base pairs is observed. This likely arises from the PB2 lid domain not yet having managed to completely clamp down and enforce strand separation compared to the late elongation structure presented below. Details of the interactions of the RNA within the active site cavity are shown in Supplementary Fig. 8c. The CpNHpp is bound in a non-catalytic configuration with only one magnesium ion. Although a capped primer was used to initiate the elongation reaction, the cap itself is not observed and the putative cap-binding site remains autoinhibited by PB2/R217. However, density tentatively assigned to primer nucleotides 4-6 is observed in the presumed product exit channel at the interface of PB1 and PB2, with A6 stacking on PB2/H305 (Supplementary Fig. 8a–c). In the absence of a priming loop, TiLV polymerase template exit tunnel always remains open, different from FluPol (*Articulavirales*) or LACV-L (*Bunyavirales*) in which the RNA duplex extension induces the extrusion of the priming loop or the template exit plug, respectively[27,31] (Supplementary Fig. 9). Overall, compared to the initiation state, the PB1 thumb (338-454) and bridge (455-515) and PB2-N, including the loop 24-32 and the lid (77-117), rotate relative to PA and the N-terminal region of PB1, expanding the active site cavity to accommodate the product-template duplex (Fig. 6e).

The higher resolution structure with the additionally bound 3′ ends mimics a later elongation state in which the outgoing template would have translocated into the 3′ end secondary site. To confirm this template trajectory, we determined a late elongation structure at 3.1 Å resolution (Table 1, Supplementary Table 1c, Supplementary Note 5) using the v40-S template (Fig. 6d, right). Upon addition of the capped 13-mer primer, ATP, UTP, CTP, Mg2+ and GpCpp, the primer is now extended by 19 nts to a 32-mer before stalling with GpCpp base-pairing with template nucleotide C19 (Fig. 6d, right), as demonstrated in the in vitro activity assay (Fig. 7c, lane 7). The template has now translocated enough to fill the template exit channel and dock in the secondary site as shown by the clear tube of density linking the top of the distal duplex to well-defined density for the template 3′ end in the secondary site (Supplementary Fig. 8d, e). Furthermore, the product-template duplex is now the expected length with the helical lid moving slightly so that PB2/D102 and G106 pack against the tenth base pair. There is also a curved tube of density connecting the template in the active site to the 5′ hook (Supplementary Fig. 8e), showing that a few more nucleotides could be incorporated before the hook becomes under tension. Apart from this, the protein structure is very similar between the early and late elongation states. Overall, these structures show that the template trajectory first described for FluPol[32], which facilitates template recycling for further rounds of

transcription/replication, is a conserved feature of all sNSV polymerases (Supplementary Fig. 9).

## Conformational flexibility of TiLV polymerase

For most functional states described above, additional cryo-EM structures were determined that exhibit different positons of some of the flexible linked domains (Table 1). Of particular interest is the observation of a replicase-like configuration of TiLV polymerase in which the ENDO-like and PB2-C domains are rearranged as previously described for FluPol (Fig. 1d, Supplementary Fig. 10a, b)[21–24]. This structure occurred as a subpopulation in the elongation state sample (Supplementary Note 4) and has the vRNA promoter bound in mode A and initiating CpNHpp is at the +1 position base pairing with template nucleotide G2. The replicase conformation is so-called as it occurs in an asymmetric dimer with the encapsidase in the presumed influenza C replication complex[24]. In the transcriptase conformation, the PA endonuclease faces the PB2 cap-binding domain, in a configuration compatible with cap-snatching (Supplementary Fig. 10c), with the PB2 627 and PB2 NLS domains contacting each other and interacting with PB1 (Supplementary Fig. 1d). To switch from the transcriptase conformation to the replicase conformation, the TiLV ENDO-like domain rotates by ~160 degrees and the PB2 NLS domain separates from the 627 domain (which remains flexible) to pack against the ENDO-like domain, interacting through hydrophobic and electrostatic interactions (Supplementary Fig. 10c). Concomitantly, the PB2 CBD-like and midlink domains rotate by ~180 degrees to pack against TiLV PB2-N and PB1 (Supplementary Fig. 10c). The C-terminal extension to the folded part of the TiLV NLS domain (i.e., beyond residue 420), which actually bears the putative NLS, is not observed in any of the TiLV structures. In contrast, in the FluPol replicase conformation, the corresponding region forms an alpha helix interacting with the endonuclease domain (Fig. 1e)[21–23]. In some of the TiLV cryo-EM structures the full replicase configuration is not observed, only the rotated endonuclease with or without associated NLS domain (Table 1), highlighting the flexible nature of these peripheral domains. Finally, we note that for the transcriptase conformation, there is a systematic difference between complete polymerase structures (i.e., including ENDO-like and PB2-C domains) and those of the core only. Packing of the 627-like domain against PB1 induces a conformational change that propagates towards the active site (Supplementary Fig. 10e), although the significance of this is unclear.

## Discussion

In this study, we structurally and functionally characterized the heterotrimeric TiLV polymerase in complex with both the vRNA and cRNA promoters. We thus report the first complete structure of a polymerase from the *Amnoonviridae*, an evolutionary distant viral family from the *Orthomyxoviridae* (which encompasses Influenza, Thogoto, Quaranja and Isa-viruses), both within the *Articulavirales* order. Recently, metagenomics[33] has revealed a number of other *Amnoonviridae* species, mainly from fish, including Flavolineata, Namensis, Asotus1, Asotus2, Przewalskii and Stewartii viruses[34,35]. Sequence alignments of the putative polymerase segments (full-length sequences only, Supplementary Notes 6–9), shows, as expected, reasonable

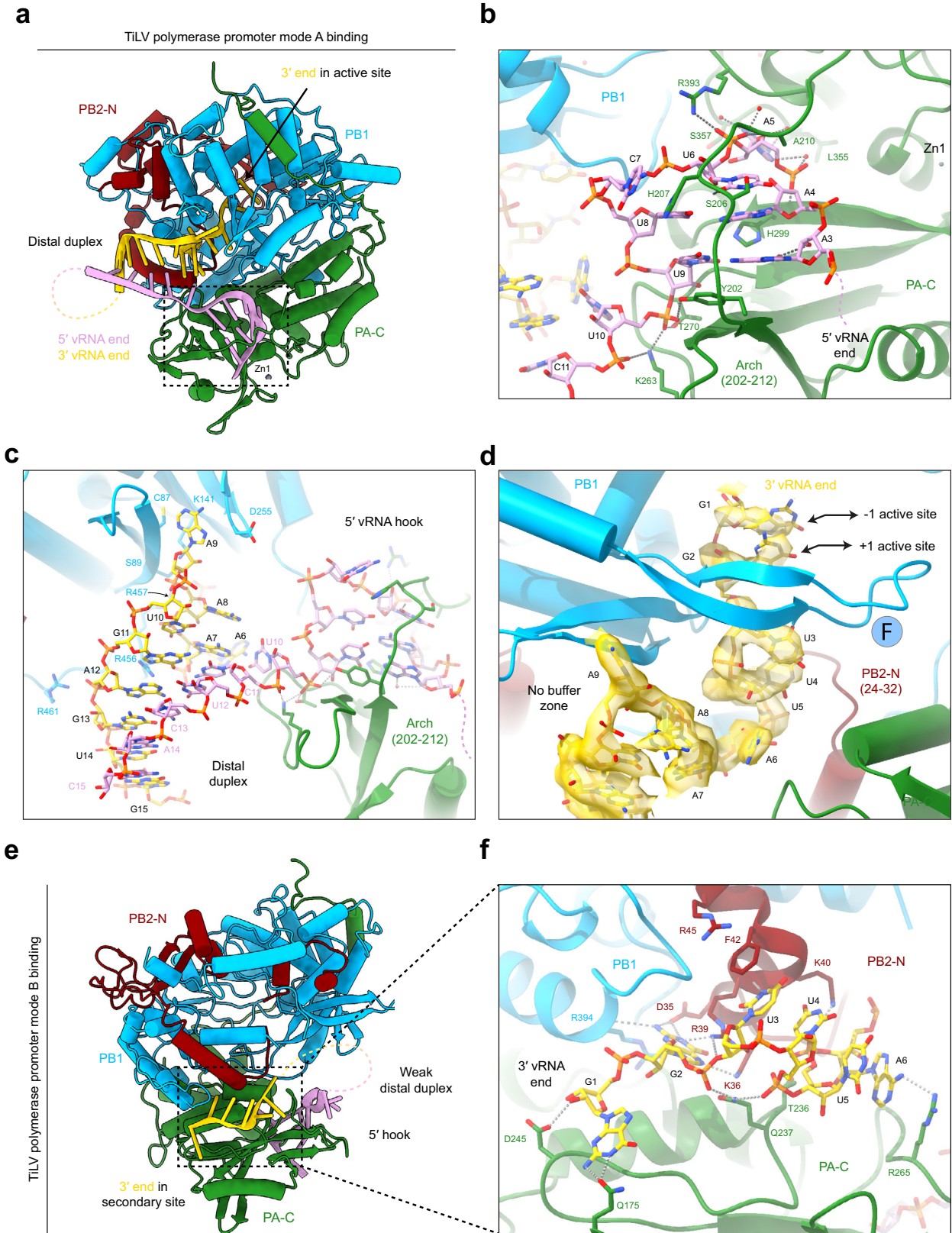

conservation of the PB1 subunit, but significant divergence in the PB2 and PA segments, indicating sparse sampling of this family to date[35]. For PB1, sequences from Tilapia Lake, Flavolineata and Namensis viruses are shorter (~520 aa) compared to those from Asotus1, Asotus2, Przewalskii and Stewartii viruses (~567 aa), which have ~24 residue insertions in both the fingers and thumb (highlighted in blue,

Supplementary Note 6). For PB2, TiLV and Namensis have significant homology, with zinc binding residues and most but not all putative cap-binding or inhibiting residues conserved (Supplementary Note 7), whereas the other four (Asotus1, Asotus2, Przewalskii and Stewartii) form a divergent clade, which the Zn2 site is conserved but not Zn3 and the putative NLS sequence is only apparent in TiLV (Supplementary

**Fig. 5 | TiLV polymerase binding to the vRNA promoter. a** Cartoon representation of TiLV polymerase bound to the 40-mer vRNA loop in promoter mode A. PA is coloured in green, PB1 in light blue and PB2-N in dark red. The 5′ vRNA end is in pink, the 3′ vRNA end in gold. The 3′ end is directed towards the polymerase active site. The 5′/3′ ends form a distal duplex with the flexible linker nucleotides shown as a dotted line. **b** Close-up view of the bound 5′ vRNA hook. TiLV polymerase subunits are coloured as in (**a**). The two first 5′ end nucleotides are flexible and represented as a dotted line. The PA arch (202-212) stabilizing the 5′ hook structure is indicated. Water molecules are red spheres and hydrogen bonds grey dotted lines. Key interacting residues are shown as sticks. **c** Close-up view of the distal duplex formed between the 5′ and 3′ vRNA ends with annotation as in (**b**). 3′ end nucleotides are numbered from the 3′ end, with nucleotides 1-5 omitted for clarity. **d** Close-up view of the vRNA 3′ end entering the TiLV polymerase active site in pre-initiation state mode A, with annotation as in **c**. The RdRp motif F β-hairpin is shown, as well as the −1/+1 active site positions occupied respectively by G1 and G2. **e** Cartoon representation of TiLV polymerase core bound to the 40-mer vRNA loop in promoter mode B with annotation as in (**a**). The 3′ end is bound to the polymerase surface in the so-called "secondary site". The 5′/3′ ends form a weak distal duplex in the closed core conformation. **f** Close-up view of the 3′ vRNA end bound in the secondary site with key interacting residues shown as sticks. Annotation and colours as in (**b**) and (**c**).

Note 8). For PA, there is very low overall homology between the three full-length sequences (TiLV, Asotus1, Flavolineata), with the Zn1 binding residues being conserved but not the putative endonuclease 'active' site residues (Supplementary Note 9). Most recently, metagenomics of corals in addition to transcriptome mining has led to identification of a proposed new family, *Cnidenomoviridae*, within the *Articulavirales* order, which would now include at least four families: *Orthomyxoviridae*, *Amnoonviridae*, *Quaranjaviridae*, and *Cnidenomoviridae*[36]. Future analysis of additional sequences will undoubtedly shed more light on the evolution of *Articulavirales* polymerases.

To date, TiLV polymerase is the smallest known viral polymerase from the sNSV group with a molecular weight of only 155 kDa, around 60% the size of FluPol or *Bunyavirales* L proteins. The putative viral nucleoprotein (segment 4) is also considerably smaller than its influenza counterpart[13]. Remarkably, the small polymerase size is achieved by an overall pruning of each subunit whilst maintaining all the functional domains typically found in orthomyxo-like viral polymerases. This is achieved in a variety of ways as described above (Figs. 2–4). A small polymerase size (and small size of gene segments in general, at least for TiLV) seems to be a feature of the *Amnoonviridae*. Given that bunya- and orthomyxo-viruses have similar sized, large polymerases, yet are probably more distantly related, it suggests that some evolutionary selective pressure may have led to a general gene size reduction in *Amnoonviridae*, perhaps the need to package ten distinct proteins/segments in the case of TiLV. Nevertheless, the minimal TiLV heterotrimeric complex can take up active transcriptase- or replicase-like conformations, equivalent to those found for FluPol. However, we find that the TiLV ENDO-like domain does not have an obvious metal-binding active site and is unable to cleave RNA in vitro (Fig. 3). Similarly, the TiLV CBD-like domain does not bind m⁷GTP in vitro, which could be explained by consistent autoinhibited by PB2/R217 (Fig. 4). However, these anomalies are shared with some other sNSV viral polymerases (Supplementary Fig. 2, 3). For example, the isolated Thogoto and Dhori virus PA-ENDO like domains do not exhibit any catalytic residues and are inactive in vitro and the Thogoto PB2 CBD-like domain is autoinhibited and does not bind cap in vitro[25]. Similarly, the Lassa virus CBD-like domain does not bind cap in vitro[37,38]. On the other hand, the SFTSV-L CBD undergoes a conformational change that releases auto-inhibition to enable 5′-cap binding and cap-dependent transcription[29] (Supplementary Fig. 3). Whether or not, TILV polymerase performs cap-snatching and if so, how, remains to be elucidated.

Like all sNSV polymerases, TiLV polymerase binds the 5′ end promoter region of vRNA or cRNA as a hook, although it is exceptionally small (nucleotides 3-9) with the first two 5′ end nucleotides overhanging. The 3′ end (vRNA and cRNA) can bind in the RdRp active site (mode A) or in the secondary site the surface of TiLV polymerase core (mode B) (Fig. 5). Indeed, the late elongation structure shows that the trajectory of the translocating template and is as first described for FluPol (but also common to *Bunyavirales*), suggesting that the mechanism of template recycling is likely also similar[32]. However, there is no conserved oligo-uridine sequence close to the 5′ end (Fig. 1b) that

might act as a polyadenylation signal, as in influenza virus. The mechanisms of transcription termination and potential polyadenylation therefore still need to be elucidated. Finally, it is interesting to note that there is no Watson-Crick complementarity between the first position of the 5′ and 3′ ends (G:G, C:C for vRNA and cRNA, respectively, Fig. 1b). Perhaps this allows escape from anti-viral innate immune receptors such as RIG-I that depend on recognition of a canonical blunt-end 5′ppp-dsRNA[39].

In vitro assays and structural data on initiation, early and late elongation states show that promoter-bound, recombinant TiLV polymerase is able to perform primed RNA synthesis with primers ending in …CC-3′ (Fig. 6; Fig. 7) and that the mechanism of strand separation and the template trajectory is as described for FluPol. However, unprimed synthesis was not observed in biochemical assays despite structures showing both 3′ vRNA and cRNA ends aligned in the active site consistent with terminal initiation, with, in the case of vRNA, CTP at the +1 position base-pairing with G2 of the template. This is perhaps due to the absence of a priming loop, which is required to stabilise the second incoming nucleotide at the −1 position during terminal vRNA to cRNA initiation by FluPol. Since for TiLV, the 3′ ends of vRNA and cRNA are of the same length and position similarly in the active site, TiLV cRNA to vRNA replication may be simpler than for FluPol, where cRNA to vRNA synthesis initiates internally, requiring a potential trans-activating apo-polymerase for template realignment[23]. For bunyavirus polymerase, replication initiation relies heavily on prime-and-realign[18].

Overall, our results provide the first structural and functional data on a prototypic viral polymerase from the *Amnoonviridae* family that show functionality is not compromised despite an unprecedented reduction in size. Unfortunately, the current lack of reverse genetics or a minireplicon system for TiLV limits the extent to which viral RNA synthesis mechanisms can be studied and tested in cells. However, our highlighting of conserved and idiosyncratic features compared to classical orthomyxoviruses will contribute to establishing such systems, pave the way for a better understanding of TiLV replication and perhaps help combat this emerging pathogen of tilapia fish, an important human food resource.

## Methods

### Cloning, expression and purification of the TiLV polymerase heterotrimeric complex

The 10 TiLV open reading frames, codon optimized for insect cells expression, were ordered at Genscript and sub cloned into multiple pFastBac Dual vectors using EcoRI and SpeI (pFastBac Dual SEGX, with X being the segment number). Plasmids containing each single segment were PCR amplified using primers sequences 5′-CGCTGAGCAA-TAACTATCATAACCCCTAGGAGATCCGAACCAGATAAGTG-3′ and 5′-GGTTATGATAGTTATTGCTCAGCGCTCAAGCAGTGATCAGATCCAGA-CATG-3′, re-circularized by Gibson cloning thus creating 10 pFastBac plasmids compatible with the biGBac cloning system[40]. Single cassettes encoding for the 10 TiLV segments were amplified using the biGBAc standard primers and assembled by Gibson assembly into psBIG1a and psBIG1b. psBIG1a and psBIG1b were linearized with Swa1,

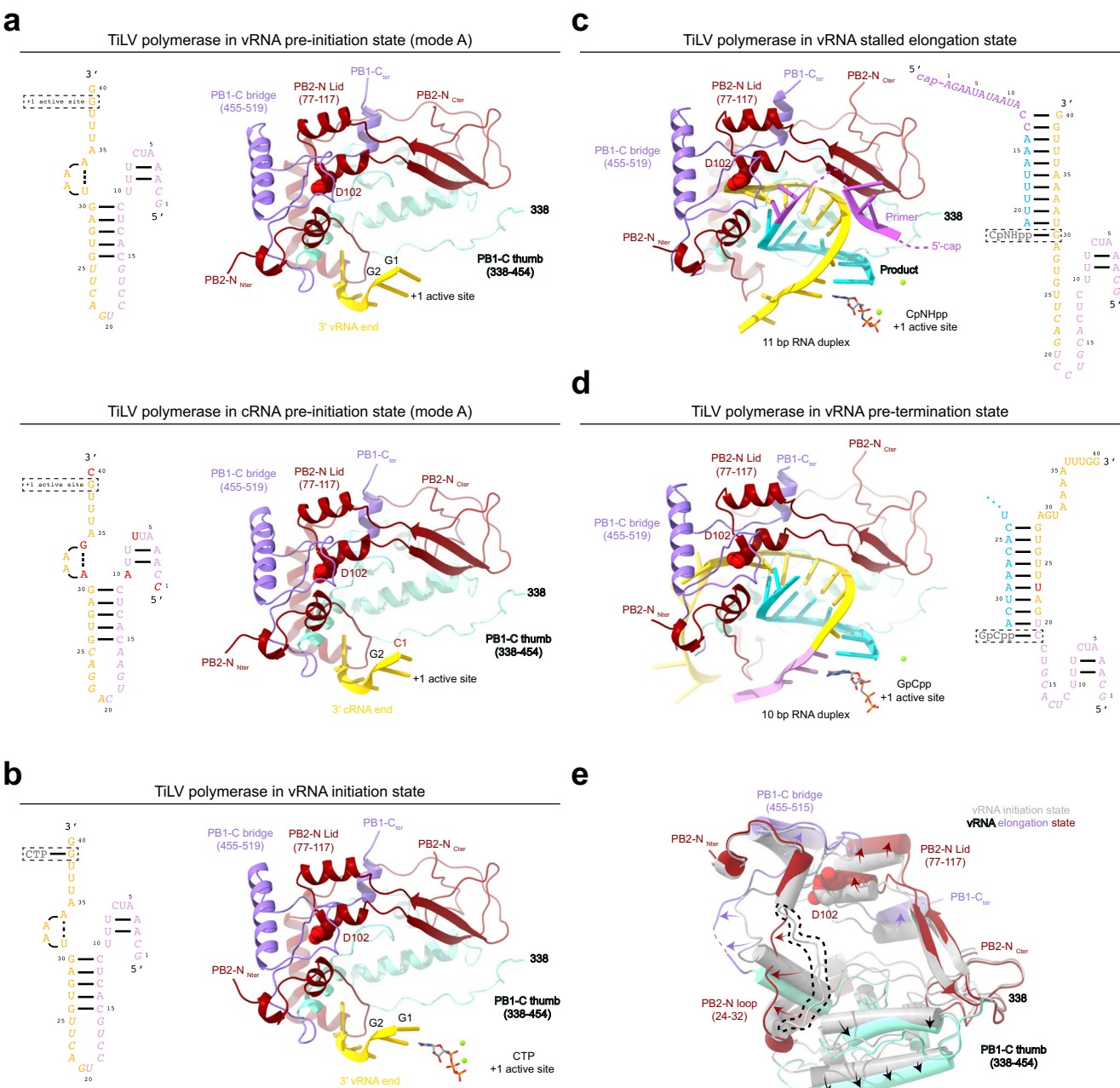

**Fig. 6 | Structural snapshots of TiLV polymerase from pre-initiation to pre-termination.** **a** TiLV polymerase in vRNA and cRNA pre-initiation state mode A. Top: Close-up of the 3′ vRNA end in the TiLV polymerase active site. The 3′ end is coloured in gold and nucleotides are numbered from the 3′ to the 5′, with G1 and G2 being in, respectively, the −1/+1 active site positions. PB1-C bridge domain, PB1-C thumb domain and PB2-N are in purple, aquamarine and dark red respectively. PB2-N/D102 from the lid domain is in spheres. Left: Schematic of the overall vRNA conformation with the +1 position within a dotted rectangle. Flexible nucleotides are in italics. Bottom: Close-up of the 3′ cRNA end in the TiLV polymerase active site, annotated as in a, except that C1 and G2 are, respectively, in the −1/+1 positions. Left: Schematic of the overall cRNA conformation annotated as in above. Nucleotide differences between the vRNA and the cRNA are highlighted in red. **b** TiLV polymerase in the vRNA initiation state with incoming CTP aligning with G2, annotated as in a. Magnesium ions (Mg2+) are shown as green spheres. Left: Schematic of the overall RNA conformation, annotated as in a, with CTP at the +1 position within a dotted rectangle. **c** TiLV polymerase stalled in early elongation state with an internal 11 base pair (bp) product-template RNA duplex. Annotations as in a. The 3′ vRNA end (gold) is not yet pushed back by PB2-N D102 lid residue,

shown as spheres. The 13-mer capped primer ending by...CC-3′ is in magenta and the incorporated nucleotides in cyan. A CpNHpp is in the +1 active site position. Magnesium ions (Mg2+) are shown as green spheres. Right: Schematic of the overall RNA conformation with colours as above. **d** TiLV polymerase stalled in a late elongation, pre-termination state showing the canonical 10 base pair product-template RNA duplex, annotated as in **a**. After strand separation (enforced by PB2-N D102, shown as spheres) the template (gold) continues through the exit channel into the secondary 3′ end binding site. The 13-mer capped primer ending by...CC-3′ is no longer visible. Incorporated nucleotides are in cyan and GpCpp is in the +1 position. Right: Schematic of the overall RNA conformation with the v40-S C23U mutation highlighted in red. The GpCpp at the +1 position is within a dotted rectangle. Flexible nucleotides are in italics. **e** TiLV polymerase movements upon product elongation. The vRNA initiation state (grey) was superposed via PB1 with the early-elongation state (colours). The RNA is omitted for clarity. Only significant movements of PB1-C and PB2-N elements are shown (arrows). The PB2-N loop (24-32) in the vRNA initiation state, which has to displace to allow elongation, is surrounded by a dotted line.

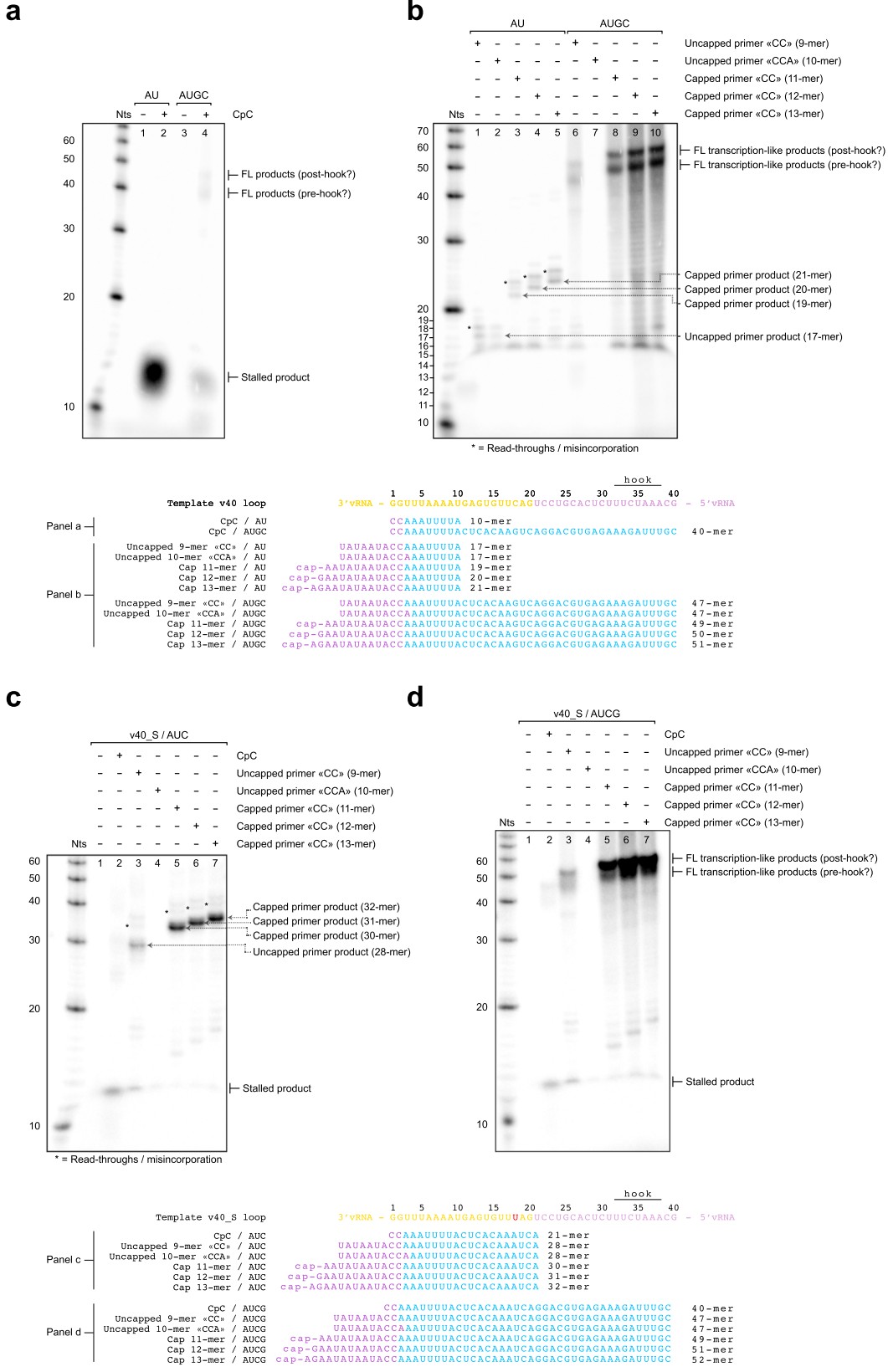

resulting in psBIG1a TiLV 1-3-5-7-9 and psBIG1b TiLV 2-4-6-8-10. Finally, psBIG1a TiLV 1-3-5-7-9 and psBIG1b TiLV 2-4-6-8-10 were digested using Pme1 and fragments encoding the 10 TiLV proteins were cloned by Gibson assembly into psBIG2ab, linearized using Pme1, to finally give psBIG2ab TiLV-10SEG.

The construct encoding for segment 1, segment 2 (with a Tobacco Etch Virus (TEV) protease cleavage site followed by a 10x-histidine tag on the C-terminus) and segment 3 proteins was obtained as following:
1.  A pFastBac Dual compatible with the biGBac system was digested with EcoR1 and Spe1 as well as the coding sequence of segment 1,

**Fig. 7 | TiLV polymerase in vitro RNA synthesis assays. a** Urea-PAGE analysis of in vitro TiLV polymerase activity using the 40-mer vRNA loop (v40) as template, without (lane 1, 3) or with (lane 2, 4) CpC dinucleotide, using ATP and UTP only (AU) (lane 1–2) or all NTPs (AUGC) (lane 3-4). Stalled product and putative full-length (FL) products are indicated on the right (post or pre-hook?). The decade marker (Nts) is on the left. Theoretical products are shown at the bottom of the gel. The 5′/3′ ends of the template are respectively coloured in plum and gold, the primer in magenta, the theoretically incorporated nucleotides in blue. Source data are provided as a Source Data file. (n = 6 independent experiments). **b** Urea-PAGE analysis of in vitro TiLV polymerase transcription-like activity using the v40 loop as template, ATP and UTP (AU) (lane 1-5) or all NTPs (AUGC) (lane 6-10), uncapped primers ending by...CC-3′ (lane 1, 6) or...CCA-3′ (lane 2, 7), capped primers ending by...CC-3′ of different lengths (11-mer, lane 3, 8; 12-mer, lane 4, 9; 13-mer, lane 5, 10). Expected stalled and full-length transcription-like products are indicated on the right. Read-throughs due to misincorporation are indicated with a star. Theoretical products are shown as in (**a**). Source data are provided as a Source Data file

(n = 6 independent experiments). **c** Urea-PAGE analysis of in vitro TiLV polymerase transcription-like activity using the v40-S loop with C23U mutation as template, ATP, UTP and CTP (AUC) (lane 1), CpC (lane 2), uncapped primers ending by...CC-3′ (lane 3) or...CCA-3′ (lane 4), capped primers ending by...CC-3′ of different lengths (11-mer, lane 5; 12-mer, lane 6; 13-mer, lane 7). Expected stalled products are indicated on the right. Read-throughs due to misincorporation are indicated with a star. Theoretical products are shown as in (**a**). The C23U mutation is coloured in red. Source data are provided as a Source Data file (n = 3 independent experiments). **d** Urea-PAGE analysis of in vitro TiLV polymerase transcription-like activity using the v40-S loop with C23U mutation as template, all NTPs (AUCG) (lane 1), CpC (lane 2), uncapped primers ending by...CC-3′ (lane 3) or...CCA-3′ (lane 4), capped primers ending by...CC-3′ of different lengths (11-mer, lane 5; 12-mer, lane 6; 13-mer, lane 7). Expected stalled products are indicated on the right with. Read-throughs due to misincorporation are indicated with a star. Theoretical products are shown as in (**a**). Source data are provided as a Source Data file (n = 3 independent experiments).

PCR amplified using primers 5′-ATACCGTCCCACCATCGGGC-3′ and 5′-CGCGACTAGTTTAGCATCCAGCAGTA GGCTGGAC-3′. Insert and backbone were ligated using T4 DNA ligase (NEB) following manufacturer recommendations giving the pFastBac Dual TiLV-SEG1-noHis.

2. Segment 2, PCR amplified using primers 5′-ATACCGTCCCACC ATCGGGC-3′ and 5′-GCGACTAGTTTAGTGGTGGTGGTGATGAT GGTGATGATGGTGTCCCTGGAAGTACAGGTTTTCTGAGCCAGAA CCCTGGTTCAGGTCCATGTTTTCAGC-3′, was digested using EcoR1 and Spe1 for subsequent ligation into a pFastBac Dual opened with EcoR1 and Spe1 using the T4 DNA ligase (NEB) following manufacturer recommendations, giving the pFastBac Dual TiLV-SEG2-TEV-His.

3. Finally, cassettes encoding for TiLV-SEG1-noHis, TiLV-SEG2-TEV-His and TiLV-SEG3 were amplified using standard biGBac primers and assembled by Gibson cloning into psBIG1a, opened by Swa1, and obtain psBIG1a TiLV-SEG1-noHis_SEG2-TEV-His_SEG3.

EMBacY bacmids[41] containing the TiLV polymerase expression plasmid were generated and used for insect cells expression. For large-scale expression, *Trichoplusia ni* High 5 cells (ThermoFisher) at $0.8-1 \times 10^6$ cells/mL concentration were infected by adding 1% of virus. Expression was stopped 72 to 96 h after the day of proliferation arrest and cells were harvested by centrifugation (1000 g, 20 min, 4 °C). The cells were disrupted by sonication for 4 min (5 s ON, 20 s OFF, 40% amplitude) on ice in lysis buffer (50 mM HEPES pH 8, 500 mM NaCl, 20 mM imidazole, 0.5 mM TCEP and 10 % glycerol) with cOmplete EDTA-free Protease Inhibitor Cocktail (Roche). After lysate centrifugation at 48,384 g for 45 min at 4 °C, the soluble fraction was loaded on a HisTrap HP ion affinity chromatography (Cytiva). Bound proteins were subjected to two sequential washes steps using (i) the lysis buffer supplemented by 1 M NaCl and (ii) the lysis buffer supplemented by 50 mM imidazole. Bound proteins were eluted using initial lysis buffer supplemented by 500 mM imidazole. TiLV heterotrimeric complex fractions were pooled and diluted to reach the heparin-loading buffer concentration (50 mM HEPES pH 8, 250 mM NaCl, 0.5 mM TCEP, 5% glycerol). Proteins were loaded on a HiTrap Heparin HP (Cytiva) column, washed using the heparin-loading buffer and eluted using 50 mM HEPES pH 8, 1 M NaCl, 2 mM TCEP, 5% glycerol. The nucleic acid free TiLV heterotrimeric complex fractions (ratio $A_{260/280} = 0.6$) were then concentrated using Amicon Ultra (30 kDa cut-off), flash frozen in liquid nitrogen, and stored at −80 C for further use.

### Samples for cryoEM

**TiLV polymerase in vRNA pre-initiation state (sample 1).** To trap TiLV polymerase bound to the 5′/3′ vRNA promoters, 1.6 μM of TiLV polymerase were mixed with 4.8 μM of the TiLV vRNA loop in the cryo-EM buffer and incubated for 1 h at 4 °C. Before proceeding to grid freezing, the sample was centrifuged for 5 min, 11,000 g and kept at 4 C.

**TiLV polymerase in cRNA pre-initiation state (sample 2).** To trap TiLV polymerase bound to the 5′/3′ cRNA promoters, 1.6 μM of TiLV polymerase were mixed with 4.8 μM of the TiLV cRNA loop in the cryo-EM buffer and incubated for 1 h at 4 °C. Before proceeding to grid freezing, the sample was centrifuged for 5 min, 11,000 g and kept at 4 C.

**TiLV polymerase in vRNA initiation state (sample 3).** To trap TiLV polymerase in the vRNA initiation state, 1.6 μM of TiLV polymerase were mixed with an equimolar amount of TiLV vRNA loop in a cryo-EM buffer (50 mM HEPES pH 8, 150 mM NaCl, 2 mM TCEP) supplemented with 100 μM CTP and 10 mM MgCl₂, incubated for 1 h at 4 °C. Before proceeding to grid freezing, the sample was centrifuged for 5 min, 11,000 g and kept at 4 C.

**TiLV polymerase in elongation state (sample 4).** To trap TiLV polymerase in an early-elongation state, 1.6 μM of TiLV polymerase were mixed with 1.3 μM of the TiLV vRNA loop and 16 μM of capped RNA primer 13-mer in the cryo-EM buffer supplemented with 100 μM ATP, 100 μM UTP, 100 μM CpNHpp (Jena Bioscience) and 10 mM MgCl₂. The mix was incubated for 4 h at 24 C. Before proceeding to grid freezing, the sample was centrifuged for 5 min, 11000 g and kept at 4 C.

**TiLV polymerase in pre-termination state (sample 5).** To trap TiLV polymerase in a late elongation, pre-termination state, 1.6 μM of TiLV polymerase were mixed with 1.3 μM of the TiLV vRNA 40-S loop (C23U) and 16 μM of capped RNA primer 13-mer in the cryo-EM buffer supplemented with 100 μM ATP, 100 μM UTP, 100 μM CTP, 100 μM GpCpp (Jena Bioscience) and 10 mM MgCl₂. The mix was incubated for 4 h at 24 C. Before proceeding to grids freezing, the sample was centrifuged for 5 min, 11,000 g and kept at 4 C.

### CryoEM grid preparation and data collection

For grid preparation, 1.5 μl of sample was applied on each sides of glow discharged (PELCO easiGlow™ Glow Discharge Cleaning System: 45 s, 30 mA, 0.45 mBar) or plasma cleaned (Fischione 1070 Plasma Cleaner: 1 min 10, 90% oxygen, 10% argon) grids (UltrAufoil 1.2/1.3, Au 300). Excess solution was blotted for 3 to 5 s, blot force 0, 100% humidity at 4 °C with a Vitrobot Mark IV (ThermoFisher) before plunge freezing in liquid ethane.

Automated data collection of the sample 1 was performed on a TEM Glacios microscope (ThermoFisher) operated at 200 kV equipped with a K2 direct electron detector camera (Gatan) using SerialEM[42]. Coma and astigmatism correction were performed on a carbon grid. Movies of 40 frames were recorded in counting mode at a ×36,000

magnification giving a pixel size of 1.1 Å with defocus ranging from −0.8 to −2.0 μm. Total exposure dose was ~40 e⁻/Å².

Automated data collection of the sample 2, 3, 4 were performed on a TEM Titan Krios G3 (Thermo Fisher) operated at 300 kV equipped with a K3 (Gatan) direct electron detector camera and a BioQuantum energy filter, using EPU. Coma and astigmatism correction were performed on a carbon grid. Micrographs were recorded in counting mode at a ×105,000 magnification giving a pixel size of 0.84 Å with defocus ranging from −0.8 to −2.0 μm. For sample 2 and 3, gain-normalized movies of 50 frames were collected with a total exposure of ~50 e⁻/Å². For sample 4, gain-normalized movies of 40 frames were collected with a total exposure of ~40 e⁻/Å².

Automated data collection of the sample 5 was performed on a TEM Glacios (Thermo Fisher) operated at 200 kV equipped with an F4i (ThermoFisher) direct electron detector camera and a SelectrisX energy filter, using EPU. Coma and astigmatism corrections were performed on a carbon grid. Micrographs were recorded in counting mode at a ×130,000 magnification giving a pixel size of 0.878 Å with defocus ranging from −0.8 to −2.0 μm. EER movies were collected with a total exposure of ~40 e⁻/Å².

## Image processing

For each collected dataset, movie drift correction was performed using Relion's Motioncor implementation, with 5 × 5 patch for K2/F4i movies and 7 × 5 patch for K3 movies, using all movie frames[43]. For images collected on Glacios cryo-TEM (ThermoFisher), both gain reference and camera defect corrections were applied. All additional initial image-processing steps were performed in cryoSPARC v3.3 or v4.2.1[44]. CTF parameters were determined using "Patch CTF estimation", realigned micrographs were then inspected and low-quality images were manually discarded. To obtain an initial 3D reconstruction of TiLV polymerase, particles were picked on few hundreds micrographs using a circular blob with a diameter ranging from 90 to 140 Å. Particles were extracted, 2D classified and subjected to an "ab-initio reconstruction" job. The best initial model was further 3D refined and used to prepare 2D templates. For each dataset, particles were then picked using the template picker and extracted from dose-weighted micrographs using a box size of 220 × 220 pixels² for the "Glacios / K2" dataset, 300 × 300 pixels² for the "Titan Krios" datasets, and 340 × 340 pixels² for the "Glacios / F4i - SelectrisX" dataset.

The same image processing approach was used to separate the different TiLV polymerase conformation. Successive 2D classifications were used to eliminate particles displaying poor structural features. All remaining particles were then transferred to Relion 4.0. For each dataset, particles were divided in subset of 300k to 500k particles and subjected to multiple 3D classification with coarse image-alignment sampling using a circular mask of 140 Å. For each similar TiLV polymerase conformation, particles were grouped and subjected to masked 3D refinement further followed by multiple 3D classification without alignment or using local angular searches. Once particles were properly classified, Bayesian polishing was performed and shiny particles subjected to a last masked 3D refinement. Post-processing was performed in Relion using an automatically or manually determined B-factor. For each final map, reported global resolution is based on the FSC 0.143 cut-off criteria and local resolution variations were estimated in Relion.

For detailed image processing information regarding each data collection, please refer to Supplementary Notes 1–5.

## Model building and refinement

Model building of the full-length TiLV polymerase and associated RNAs was performed de novo in COOT[45]. The Robetta[12] prediction of the structure of the polymerase core of segment 1 was used as a starting point. Models were refined using Phenix real-space refinement[46] with Ramachandran restraints. Atomic model validation was performed

using Molprobity[47] as implemented in Phenix. Model resolution according to cryo-EM map was estimated at the 0.5 FSC cutoff. Figures were generated using ChimeraX[48]. Electrostatic potential was calculated using PDB2PQR[49] and APBS[50].

## Endonuclease cleavage assay

Synthetic RNAs (Integrated DNA technologies) (i) TiLV 5′ vRNA end 1-15 (5′ - pGCA AAU CUU UCU CAC − 3′) and (ii) TiLV 3′ vRNA end 1-15 (5′ - GUG AGU AAA AUU UGG − 3′) promoters were mixed with 1 μM TiLV polymerase in a 1:1:1 molar ratio. The nuclease activity of TiLV PA-ENDO-like domain was then tested by incubating TiLV polymerase bound or not to the 5′/3′ vRNA promoters with a 41-mer U-rich RNA (5′ − GGC CAU CCU GUU UUU UUU CCC UUU UUU UUU UUC UUU UUU UU) at 2 μM in presence of multiple divalent metal ions (Mn²⁺, Mg²⁺, Zn²⁺, Co²⁺, Ca²⁺) at 1 mM final concentration in a final reaction buffer composed of 50 mM HEPES pH 8, 150 mM NaCl, 2 mM TCEP. Reactions were realized at 24 C for 1 h and stopped by adding 2x RNA loading buffer (95% formamide, 1 mM EDTA, 0.025% SDS, 0.025% bromophenol blue, 0.01% xylene cyanol). Samples were heated 5 min at 95 C and analysed on a 7 M urea 15% acrylamide gel further stained with SYBR Gold (ThermoFisher). As a positive control, a similar reaction was setup with the A/H7N9 influenza polymerase activated using both 5′ vRNA 1-14 (5′-pAGU AGA AAC AAG GC-3′) and 3′ vRNA 1-6 (5′ −UCU GCU-3′), in the same ratio as TiLV polymerase reactions, using 1 mM Mn²⁺ in a buffer containing 50 mM HEPES pH 8, 150 mM NaCl, 2 mM TCEP. Reaction was done at 30 °C for 1 h and stopped by adding 2x RNA loading buffer. Sample was heated 5 min at 95 C and analysed on a 7 M urea 15% acrylamide gel further stained with SYBR Gold (ThermoFisher).

The Decade Markers System (Ambion) was used as a molecular weight ladder.

## Cap-binding assay

TiLV polymerase and A/H7N9 influenza polymerase (positive control) were respectively mixed with TiLV vRNA loop and H7N9 vRNA loop (1:1 molar ratio) in a loading buffer (50 mM HEPES pH 8, 150 mM NaCl, 2 mM TCEP). Polymerases-RNA mixes were then incubated with immobilized γ-Aminophenyl-m⁷GTP (C10-spacer) beads (Jena Bioscience) for 2 h at 4 °C, washed extensively with the loading buffer and eluted with the loading buffer supplemented with 1 mM m⁷GTP (Jena Bioscience). Input, washed and eluted fractions and what remains bound to the beads were analysed on 4-20% Tris-glycine gel (ThermoFisher) stained with Commassie blue.

## In vitro polymerase activity assays

Synthetic RNAs were made by Integrated DNA technologies. Capped RNAs were made by Mathieu Noel and Francoise Debart (Institut des Biomolécules Max Mousseron, Montpellier, France).

TiLV 40-mer vRNA loop, enclosing the 20 first and 20 last nucleotides of segment 9 were used as a model vRNA (5′-pGCA AAU CUU UCU CAC GUC CUG ACU UGU GAG UAA AAU UUG G −3′).

TiLV 40-mer vRNA loop (v40-S), as the preceding but with the C23U mutation (5′-pGCA AAU CUU UCU CAC GUC CUG A**U**U UGU GAG UAA AAU UUG G −3′).

TiLV 40-mer cRNA loop, enclosing the 20 first and 20 last nucleotides of the complement of segment 9, were used as a model cRNA (5′-pCCA AAU UUU ACU CAC AAG UCA GGA CGU GAG AAA GAU UUG C −3′).

Synthetic uncapped 9-mer (5′-UAU AAU ACC −3′) and 10-mer (5′-UAU AAU ACC A − 3′) RNAs and capped 11-mer (5′- m⁷GTP − AAU AUA AUA CC −3′), 12-mer (5′- m⁷GTP − GAA UAU AAU ACC −3′) and 13-mer (5′- m⁷GTP − AGA AUA UAA UAC C −3′) RNAs ending by …CC-3′ were used as primers for in vitro activity assays.

For nucleotide incorporation activity assays, 0.8 μM TiLV polymerase were mixed with 1 μM TiLV 40-mer vRNA loop (v40 or v40-S) and 8 μM capped primers. Reactions were launched by adding all NTPs

or ATP/UTP or ATP/UTP/CTP with MgCl$_2$, α-$^{32}$P ATP at 1 μCi/μl (PerkinElmer) in a reaction buffer (50 mM HEPES pH 8, 150 mM NaCl, 2 mM TCEP, 10 mM MgCl$_2$) for 4 h at 24 C. Reaction temperature and magnesium concentration used were found by optimization. Decade Markers System (Ambion) was used as molecular weight ladder. Reactions were stopped by adding 2x RNA loading dye, heating 5 min at 95 C and immediately loaded on a denaturing 20% acrylamide 7 M urea gel. The gel was exposed on a storage phosphor screen and read with an Amersham Typhoon scanner (Cytiva).

### Reporting summary

Further information on research design is available in the Nature Portfolio Reporting Summary linked to this article.

## Data availability

The coordinates and EM maps generated in this study have been deposited in the Protein Data Bank and the Electron Microscopy Data Bank (see Table 1) under accession codes: PDB 8PSN EMD-17857 (Full vRNA initiation with CTP). PDB 8PSO EMD-17858 (Core vRNA initiation with CTP). PDB 8PSQ EMD-17860 (Core cRNA pre-initiation mode A). PDB 8PT7 EMD-17869 (Core with endo cRNA pre-initiation mode A). PDB 8PSS EMD-17861 (Closed core with endo cRNA mode B). PDB 8PSU EMD-17862 (Core vRNA pre-initiation mode A). PDB 8PSX EMD-17864 (Full vRNA elongation with CpNHpp). PDB 8PSZ EMD-17865 (Full vRNA elongation with CpNHpp and additional mode B promoter). PDB 8PT2 EMD-17866 (Full with open core vRNA mode B). PDB 8PTH EMD-17871 (Open core with rotated endo-NLS vRNA mode B). PDB 8PTJ EMD-17872 (Closed core with rotated endo-NLS vRNA mode B). PDB 8PT6 EMD-17868 (Replicase initiation with CpNHpp). PDB 8QZ8 EMD-18772 (Full vRNA pre-termination with GpCpp). Source data are provided with this paper.

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

## Acknowledgements

We especially thank Guy Schoehn and Eleftherios Zarkadas for supporting our access to the IBS Glacios. We acknowledge the European Synchrotron Radiation Facility and the Partnership for Structural Biology (PSB) for access to the Titan Krios CM01, and especially Eaazhisai Kandiah for setting up the multiple high-quality data collections. We thank Wojtek Galej, Sarah Schneider and Romain Linares for access to the EMBL Grenoble Glacios and Aymeric Peuch for support using the joint EMBL-IBS computer cluster. We thank Mathieu Noel and Francoise Debart (Institut des Biomolécules Max Mousseron, Montpellier, France) for synthesis of capped RNAs. This work used the platforms at the Grenoble Instruct-ERIC Centre (ISBG; UMS 3518 CNRS CEA-UGA-EMBL) with support from the French Infrastructure for Integrated Structural Biology (FRISBI, ANR-10-INSB-05-02) and GRAL, a project of the University Grenoble Alpes graduate school (Ecoles Universitaires de Recherche) CBH-EUR-GS (ANR-17-EURE-0003) within the Grenoble Partnership for Structural Biology. The IBS Electron Microscope facility is supported by the Auvergne Rhône-Alpes Region, the Fonds Feder, the Fondation pour la Recherche Médicale and GIS-IBiSA. The work was partly supported by ANR grant ANR-18-CE11-0028 to SC.

## Author contributions

M.P. and B.A. with the help of A.T. identified the TiLV polymerase components and purified the complex. B.A. performed all cryoEM data collection and image processing and all biochemical studies on the complex. S.C. conceived the project and built and refined structural models. B.A. and S.C. wrote the manuscript with input from MP.

## Funding

## Competing interests

The authors declare no competing interests.
