## [Peer Review File · Nature Communications]

Structural and functional analysis of the minimal orthomyxovirus-like polymerase of Tilapia Lake Virus from the highly diverged Amnoonviridae familyREVIEWER COMMENTS

Reviewer #1 (Remarks to the Author):

Tilapia Lake Virus (TiLV) is an orthomyxo-like virus that may cause a grave disease in tilapia and threaten the food security of millions of individuals. Despite exhibiting characteristics of orthomyxoviruses, nine out of its ten primary open reading frames (ORFs) do not display any noticeable sequence homology to other known ORFs. Only the ORF encoded by the largest segment (Segment 1 of the ten segments that comprise the TiLV genome) exhibits weak sequence homology to the influenza C virus PB1 subunit.

In this study, Arragain et al. have demonstrated that while the TiLV polymerase resembles the influenza virus polymerase, the two have distinct differences. The authors demonstrated that the three largest segments of the TiLV genome (1, 2, and 3) encode for the PB1, PB2, and PA-like polymerase subunits, respectively. By expressing the ten major proteins of TiLV in insect cells, they could demonstrate that PB2 and PA co-purified with tagged PB1 (or PB1 and PA with tagged PB2), suggesting the formation of a heterotrimeric complex. This notion was further strengthened by cryo-EM analyses that resulted in high-resolution (sub-3Å) TiLV polymerase complex structures composed of the three subunits. These structures captured the polymerase in three states of the viral RNA replication (pre-initiation, initiation, and elongation), including its binding to the viral RNA promoters. The authors also describe *in vitro* conditions for RNA synthesis by TiLV polymerase. This is a significant achievement considering there are no reported mini-replicon/reverse genetics systems for this pathogen.

These analyses revealed several features unique to TiLV polymerase compared to the influenza virus polymerase, including a remarkable size reduction for all three polymerase subunits (despite overall conservation of fold and organization). This size reduction highlights TiLV polymerase as the smallest known polymerase of segmented negative-stranded RNA viruses. Another intriguing finding relates to the mode of viral mRNA synthesis that may deviate from the known 'cap-snatching' model of influenza virus. It includes observations that the endonuclease domain in TiLV PA is truncated and devoid of activity in *in vitro* endonuclease activity assays. Moreover, the cap-binding activity of TiLV PB2 appears to be auto-inhibited, and TiLV polymerase does not bind *in vitro* to immobilized cap under conditions where influenza virus polymerase does. The detailed analyses in this study also point to additional interesting features of TiLV polymerase, such as the presence of three (likely stabilizing) zinc ions and the absence of a priming loop, which is essential for genome replication initiation in influenza viruses.

The authors used state-of-the-art methodologies and composed the manuscript in a detailed and interesting manner. The results are clearly presented and explained in a way that allows investigators who are not experts in structural biology to follow along. The flow is logical, and throughout the text, the authors repeatedly compare and contrast TiLV polymerase with polymerases of other viruses (e.g., influenza A virus and Thogotovirus) to convey clear and convincing conclusions. This study also lays the groundwork for new investigations, such as how TiLV polymerase interacts with the recently identified TiLV nucleoprotein to form replication complexes.

Overall, this manuscript provides ample novel data and is highly significant to the field of virus evolution, with implications for understanding the evolution of both the recently recognized Amnoonviridae family and the known Orthomyxoviridae family (both part of Articulavirales order), which together comprise a group of essential pathogens. Accordingly, I find this manuscript suitable for publication in Nature Communications.

Minor suggestions for improvement include:

Page 3- Reformatting the reference (Pulido et al, 2019, <https://doi.org/10.1016/j.aquaculture.2019.04.058>).

Page 3- "In addition, the uniqueness of the putative TiLV encoded proteins, apart from a recognisable polymerase PB1-like subunit, precludes hypotheses as to protein function based on sequence homology or structure prediction using AlphaFold9". Reference 31 can be added alongside Reference 9, as it discusses the discrepancy between AlphaFold and RoseTTAFold in predicting TiLV protein (NP) structure.

Additionally, Reference 31 describes a novel bioinformatics approach to predict the functions of TiLV proteins based on a combined feature analysis of orthomyxoviruses. This approach predicted

that Segments 1, 2, and 3 encode for the three subunits of TiLV polymerase. As the data in the current manuscript align with these predictions, it would be beneficial to mention this successful prediction of the combined feature analysis, as it should reinforce the use of this prediction tool for studying other viruses.

Page 5- In multiple instances, "segment 1 protein" should be replaced with "segment 1-encoded protein".

Page 6 – "Fig. 1de" should be corrected to "Fig. 1d,e".

Page 8 – "Despite being minimal, the TiLV PB1 subunit contains all the conserved RdRp functional motifs". The presence of the conserved RdRp motifs in TiLV PB1 was previously demonstrated (by sequence alignment) in Reference 2, and thus, it would be appropriate to add this reference to this sentence.

EXTENDED DATA FIGURE 1 – It is assumed that the agarose gel in section 'a' represents a screen (by digestion) of ten colonies for the correct psBIG2ab plasmid. If so, for clarity, this should be explained/mentioned in the legend (and not just the simulation). Otherwise, the ten lanes may be interpreted as an analysis of ten different plasmids for each of the ten segments (which is not the case).

Reviewer #2 (Remarks to the Author):

In the manuscript presented by Benoit Arragain et al., the authors determine the structures and functions of polymerase of Tilapia Lake Virus (TiLV), using cryo-electron microscopy. The structures revealed that the heterotrimeric polymerase of TiLV is composed of PB1, PB2 and PA-like subunits, whose domains resemble and can all be found in influenza polymerase, despite an overall size reduction. Multiple cryo-EM structures of TiLV polymerase in different states, illustrate the mechanism it binds the vRNA and cRNA promoters and performs RNA synthesis, with both transcriptase and replicase. The different features comparing with the polymerase of flu were also demonstrated.

Overall, it is commendable that the manuscript reports interesting structures of RNA-dependent RNA polymerase (RdRp) of a lesser known Amnoonviridae family member. Although it has truncated size and some different features due to the host and species diversity, the reported TiLV polymerase has the core features (for example, the palm, finger, thumb domains) of the RdRP which have been widely reported. The reported structures will be of some interest to the virologists together with the researchers related with biology, virology, medicine, etc.

Other comments in no particular order:

1, Authors majorly compared the structures and functions with that of the influenza polymerase. There are different types of influenza. Are the polymerases of different types of flu same? In the manuscript, the authors used two FluPols, PDB: 4WSB and 5EPI, come from influenza A and B. It may be better to be introduced and mentioned firstly and indicate the polymerase you compared with is from FluA or B.

2, Present the full name of PB (polymerase basic proteins?), PA (polymerase acid protein?), vRNA(viral RNA?) etc., when they are arisen first time in the text.

3, In fig.2, 3, and 4, the pdb of flu polymerase is 4WSB or 5EPI? Please indicate them.

4, P6, "The TiLV 5' and 3' ends are quasi-complementary over 15 nucleotides.....", add reference.

5, Fig.4 C, on right panel, add a note or marker in the image to indicate the structure is for flu PB, like the left panel to indicate the structure is for TiLV.

6, Based on the methods, authors prepared 4 samples for cryo-EM, but have submitted 12 maps to EMDB. To add a reconstruction strategy flowchart in the supplementary information, using density maps and indicate each final map with EMDB. And also to add a supplementary figure to show the Local resolution (ResMap) and FSC curve of each map.

7, P9, "In FluPol, a stable PB1-PB2 inter-subunit interaction is formed by a helical bundle involving the C-terminus.....", add a reference.

8, P17, "For PB1, TiLV, Flavolineata, Namensis are the shortest.....", change to be "PB1 of TiLV, Flavolineata, Namensis are shorter...."? All PB1 of these three viruses are composed same number of amino acids? Otherwise, which one is the shortest? Or change "shortest" to "shorter"?

Reviewer #3 (Remarks to the Author):

In this manuscript, Arragain et al. present a structural and functional characterization of the TiLV viral polymerase. TiLV is a ten segmented negative sense RNA virus that causes disease in tilapia, affecting both wild and farmed fish. The authors first identify one subunit which contains conserved motifs indicative of an RdRp domain and subsequently, through tagging/pulldown experiments, identify two other viral proteins which assemble to form a polymerase heterotrimer. This assembly is analogous to the PA, PB1, PB2 influenza virus polymerase heterotrimer.

Next, the authors use cryoEM to solve numerous structures representing stages of transcription and replication. These models are generated from high quality cryoEM maps. Functional characterization with binding experiments and in vitro assays demonstrates that under the tested reaction conditions, the polymerase does not appear to have endonuclease or cap-binding activities. In vitro activity assays with dinucleotide or cap-primed substrates demonstrate the polymerase's activity.

This is an impressive piece of work, featuring insightful biochemistry and a tour-de-force of structural biology experiments. The description of a viral polymerase RdRp from this aquatic virus provides an interesting point of comparison to the extensively studied polymerase complexes of mammalian infecting influenza viruses. The data support the conclusions, making this a strong piece of largely structural work that enables the field to further investigate functional questions. The minor weakness of the paper is the preliminary nature of the in vitro activity data as described below which presents insufficiently supported claims.

Comments

In panel 6C, CpC primed AU (lane 2) gives a large amount of product while upon inclusion of CG (lane 4) appears to potentially inhibit the polymerase. Can the authors suggest a reason for this? The mobility of the band annotated as the 10mer doesn't match with the ladder/marker and it is very diffuse. Are there multiple products present? Have these data been replicated and are they reproducible?

In panel 6C the two bands observed around the 40nt marker are described as products formed by the polymerase either reading through the 5' hook or not. Have the authors considered this may be a single RNA product that was incompletely denatured (despite boiling/urea treatment)? Perhaps the annotation of this product should be changed without additional conformation of the identity of the product.

Panel 6D, lanes 1 and 2, the text state there are additional stalling sites 2 or 4 nt longer than the expected product. The ladder suggests these products are 1 and 2 nt longer than expected. This discrepancy between the data and text should be addressed. Lane 7 is suspiciously empty, could this be a misloading? Are the stronger products observed in lanes 9 and 10 reproducible? There is no statement on how reproducible or repeatable this experiment is.

By the authors' own admission these in vitro activity data are preliminary. It may be worth removing these data or moving to supplementary/extended data. Ideally these data should be replicated.

In the cap binding assay promoter RNA was not included. Given that RNA can affect the conformation of the polymerase is it worthwhile to repeat the assay with the addition of vRNA? Similarly, would mutation of PB2/R217A relieve the block to the putative cap binding site and enable cap binding?

The influenza virus polymerase is known to interact with host proteins (Pol II-CTD and ANP32),

Can the authors speculate at all on any possible conservation of the binding sites for interaction with these proteins?

Minor points

Which influenza virus is being used for comparison to generate residue numbers and amino acid percentages? Generally, the flupol heterotrimer is ~255 kDa not 270 kDa.

How were the Zn sites identified? If it's based on the metal coordination geometry this may be worth stating in the text.

Studies on the influenza polymerase from this group and others have shown dimeric polymerase structures, were any of these observed within any of the datasets?

Panel 2D, maybe rather than annotating at "+", the positive control lane could be annotated as FluPol.

Figure 3, there is a line around some of the thumb cyan text that looks different to the others.

Figure 4, Can the authors specify the PDB of the FluPol being used?

Point by point answers to the referee's comments.

We are very thankful for the referees' positive assessment of our work and for their constructive comments and suggestions to improve the manuscript. Please note that significant changes to the main text are highlighted in cyan.

Reviewer #1 (Remarks to the Author):

Tilapia Lake Virus (TiLV) is an orthomyxo-like virus that may cause a grave disease in tilapia and threaten the food security of millions of individuals. Despite exhibiting characteristics of orthomyxoviruses, nine out of its ten primary open reading frames (ORFs) do not display any noticeable sequence homology to other known ORFs. Only the ORF encoded by the largest segment (Segment 1 of the ten segments that comprise the TiLV genome) exhibits weak sequence homology to the influenza C virus PB1 subunit.

In this study, Arragain et al. have demonstrated that while the TiLV polymerase resembles the influenza virus polymerase, the two have distinct differences. The authors demonstrated that the three largest segments of the TiLV genome (1, 2, and 3) encode for the PB1, PB2, and PA-like polymerase subunits, respectively. By expressing the ten major proteins of TiLV in insect cells, they could demonstrate that PB2 and PA co-purified with tagged PB1 (or PB1 and PA with tagged PB2), suggesting the formation of a heterotrimeric complex. This notion was further strengthened by cryo-EM analyses that resulted in high-resolution (sub-3Å) TiLV polymerase complex structures composed of the three subunits. These structures captured the polymerase in three states of the viral RNA replication (pre-initiation, initiation, and elongation), including its binding to the viral RNA promoters. The authors also describe in vitro conditions for RNA synthesis by TiLV polymerase. This is a significant achievement considering there are no reported mini-replicon/reverse genetics systems for this pathogen.

These analyses revealed several features unique to TiLV polymerase compared to the influenza virus polymerase, including a remarkable size reduction for all three polymerase subunits (despite overall conservation of fold and organization). This size reduction highlights TiLV polymerase as the smallest known polymerase of segmented negative-stranded RNA viruses. Another intriguing finding relates to the mode of viral mRNA synthesis that may deviate from the known 'cap-snatching' model of influenza virus. It includes observations that the endonuclease domain in TiLV PA is truncated and devoid of activity in in vitro endonuclease activity assays. Moreover, the cap-binding activity of TiLV PB2 appears to be auto-inhibited, and TiLV polymerase does not bind in vitro to immobilized cap under conditions where influenza virus polymerase does. The detailed analyses in this study also point to additional interesting features of TiLV polymerase, such as the presence of three (likely stabilizing) zinc ions and the absence of a priming loop, which is essential for genome replication initiation in influenza viruses.

The authors used state-of-the-art methodologies and composed the manuscript in a detailed and interesting manner. The results are clearly presented and explained in a way that allows investigators who are not experts in structural biology to follow along. The flow is logical, and throughout the text, the authors repeatedly compare and contrast TiLV polymerase with polymerases of other viruses (e.g., influenza A virus and Thogotovirus) to convey clear and

convincing conclusions. This study also lays the groundwork for new investigations, such as how TiLV polymerase interacts with the recently identified TiLV nucleoprotein to form replication complexes.

Overall, this manuscript provides ample novel data and is highly significant to the field of virus evolution, with implications for understanding the evolution of both the recently recognized Amnoonviridae family and the known Orthomyxoviridae family (both part of Articulavirales order), which together comprise a group of essential pathogens. Accordingly, I find this manuscript suitable for publication in Nature Communications.

Minor suggestions for improvement include:

Page 3- *Reformatting the reference (Pulido et al, 2019, <https://doi.org/10.1016/j.aquaculture.2019.04.058>).*

This has been done.

Page 3- *"In addition, the uniqueness of the putative TiLV encoded proteins, apart from a recognisable polymerase PB1-like subunit, precludes hypotheses as to protein function based on sequence homology or structure prediction using Alphafold9". Reference 31 can be added alongside Reference 9, as it discusses the discrepancy between Alphafold and RoseTTAFold in predicting TiLV protein (NP) structure.*

Reference 31 has been added.

Additionally, Reference 31 describes a novel bioinformatics approach to predict the functions of TiLV proteins based on a combined feature analysis of orthomyxoviruses. This approach predicted that Segments 1, 2, and 3 encode for the three subunits of TiLV polymerase. As the data in the current manuscript align with these predictions, it would be beneficial to mention this successful prediction of the combined feature analysis, as it should reinforce the use of this prediction tool for studying other viruses.

The following sentence has been added: 'However, using a bioinformatics approach based on comparative features of orthomyxoviral proteins, segment 2 and 3 proteins were predicted to be additional polymerase subunits{Abu Rass, 2022 #2129}.'

Page 5- *In multiple instances, "segment 1 protein" should be replaced with "segment 1-encoded protein".*

These changes have been made.

Page 6 – *"Fig. 1de" should be corrected to "Fig. 1d,e".*

Corrected.

Page 8 – *"Despite being minimal, the TiLV PB1 subunit contains all the conserved RdRp functional motifs". The presence of the conserved RdRp motifs in TiLV PB1 was previously demonstrated (by sequence alignment) in Reference 2, and thus, it would be appropriate to add this reference to this sentence.*

Reference 2 has been added.

SUPPLEMENTARY FIGURE 1 – It is assumed that the agarose gel in section 'a' represents a screen (by digestion) of ten colonies for the correct psBIG2ab plasmid. If so, for clarity, this should be explained/mentioned in the legend (and not just the simulation). Otherwise, the ten lanes may be interpreted as an analysis of ten different plasmids for each of the ten segments (which is not the case).

The reviewer is right. It is now clearly stated in the Supplementary Fig.1 caption that this is a screen.

Reviewer #2 (Remarks to the Author):

In the manuscript presented by Benoit Arragain et al., the authors determine the structures and functions of polymerase of Tilapia Lake Virus (TiLV), using cryo-electron microscopy. The structures revealed that the heterotrimeric polymerase of TiLV is composed of PB1, PB2 and PA-like subunits, whose domains resemble and can all be found in influenza polymerase, despite an overall size reduction. Multiple cryo-EM structures of TiLV polymerase in different states, illustrate the mechanism it binds the vRNA and cRNA promoters and performs RNA synthesis, with both transcriptase and replicase. The different features comparing with the polymerase of flu were also demonstrated.

Overall, it is commendable that the manuscript reports interesting structures of RNA-dependent RNA polymerase (RdRp) of a lesser known Amnoonviridae family member. Although it has truncated size and some different features due to the host and species diversity, the reported TiLV polymerase has the core features (for example, the palm, finger, thumb domains) of the RdRP which have been widely reported. The reported structures will be of some interest to the virologists together with the researchers related with biology, virology, medicine, etc.

Other comments in no particular order:

1, Authors majorly compared the structures and functions with that of the influenza polymerase. There are different types of influenza. Are the polymerases of different types of flu same? In the manuscript, the authors used two FluPols, PDB: 4WSB and 5EPI, come from influenza A and B.

It may be better to be introduced and mentioned firstly and indicate the polymerase you compared with is from FluA or B.

For the transcriptase conformation, we used PDB:4WSB, which is the polymerase from bat A/H17N10. For the replicase conformation, we previously used PDB:5EPI (polymerase from B/Memphis), now changed to the replicase structure from A/H5N1 (PDB:6QPF). The PDBs used are now indicated in the captions of Figures 1d and 1e respectively.

It should be noted that the overall structures and domains of all FluA and FluB polymerases are very similar at the level of detail required here.

2, *Present the full name of PB (polymerase basic proteins?), PA (polymerase acid protein?), vRNA(viral RNA?) etc., when they are arosen first time in the text.*

The abbreviations are now defined in the Introduction.

3, *In fig.2, 3, and 4, the pdb of flu polymerase is 4WSB or 5EPI? Please indicate them.*

In Fig. 2, 3, 4a, e, f, we used the polymerase structure from bat A/H17N10 (PDB: 4WSB). In Fig. 4c, we used the PB2 CBD from A/H7N9 polymerase (PDB: 7QTL). These are now indicated in the corresponding figure captions.

4, P6, *“The TiLV 5' and 3' ends are quasi-complementary over 15 nucleotides.....”*, add reference.

This observation is based on the vRNA sequences for the ten genome segments in Genbank depositions KU751814.1 to KU751823.1 as indicated in Fig. 1b and its caption. To make this clearer, this is now mentioned in the main text.

5, *Fig.4 C, on right panel, add a note or marker in the image to indicate the structure is for flu PB, like the left panel to indicate the structure is for TiLV.*

It is now indicated.

6, *Based on the methods, authors prepared 4 samples for cryo-EM, but have submitted 12 maps to EMDB. To add a reconstruction strategy flowchart in the supplementary information, using density maps and indicate each final map with EMDB. And also to add a supplementary figure to show the Local resolution (ResMap) and FSC curve of each map.*

This was already present in the Supplementary Data Figures 1-4. We are sorry this was overlooked by the referee. A Supplementary Data Figure 5 has been added for the new TiLV polymerase structure in pre-termination state.

7, P9, “In FluPol, a stable PB1-PB2 inter-subunit interaction is formed by a helical bundle involving the C-terminus.....”, add a reference.

Sugiyama et al (2009) reference added.

8, P17, “For PB1, TiLV, Flavolineata, Namensis are the shortest.....”, change to be “PB1 of TiLV, Flavolineata, Namensis are shorter....” ? All PB1 of these three viruses are composed same number of amino acids? Otherwise, which one is the shortest? Or change “shortest” to “shorter”?

The sentence has been changed to: ‘For the PB1 sequences, TiLV, Flavolineata, Namensis are shorter (~520 aa) compared to Asotus1, Asotus2, Przewalskii and Stewartii (~ 567 aa), which have ~24 residue insertions in both the fingers and thumb (highlighted in blue, now Supplementary Data 6)’.

Reviewer #3 (Remarks to the Author):

In this manuscript, Arragain et al. present a structural and functional characterization of the TiLV viral polymerase. TiLV is a ten segmented negative sense RNA virus that causes disease in tilapia, affecting both wild and farmed fish. The authors first identify one subunit which contains conserved motifs indicative of an RdRp domain and subsequently, through tagging/pulldown experiments, identify two other viral proteins which assemble to form a polymerase heterotrimer. This assembly is analogous to the PA, PB1, PB2 influenza virus polymerase heterotrimer.

Next, the authors use cryoEM to solve numerous structures representing stages of transcription and replication. These models are generated from high quality cryoEM maps. Functional characterization with binding experiments and in vitro assays demonstrates that under the tested reaction conditions, the polymerase does not appear to have endonuclease or cap-binding activities. In vitro activity assays with dinucleotide or cap-primed substrates demonstrate the polymerase's activity.

This is an impressive piece of work, featuring insightful biochemistry and a tour-de-force of structural biology experiments. The description of a viral polymerase RdRp from this aquatic virus provides an interesting point of comparison to the extensively studied polymerase complexes of mammalian infecting influenza viruses. The data support the conclusions, making this a strong piece of largely structural work that enables the field to further investigate functional questions. The minor weakness of the paper is the preliminary nature of the in vitro activity data as described below which presents insufficiently supported claims.

Comments

1. In panel 6C, CpC primed AU (lane 2) gives a large amount of product while upon inclusion of CG (lane 4) appears to potentially inhibit the polymerase. Can the authors suggest a reason for this?

As observed in the TiLV initiation state cryo-EM structure, TiLV polymerase tightly binds a CTP in the +1 active site position, but not at the -1 position. This situation has been observed in all datasets containing CTP, suggesting that a non-negligible portion of TiLV polymerase can get stalled in such a state. As there is no CTP in the reaction “CpC + AU” (now moved in the Supplementary Fig. 8a lane 2), TiLV polymerase can properly initiate and theoretically incorporate nucleotides up to position G11 from the 3' vRNA end. We hypothesise that the addition of CTP would compete with CpC priming, thereby inhibiting TiLV polymerase at the initiation level. This would explain why, upon inclusion of CTP and GTP product synthesis is severely limited (Supplementary Fig. 8a, lane 4),

2. The mobility of the band annotated as the 10mer doesn't match with the ladder/marker and it is very diffuse. Are there multiple products present?

We agree with the reviewer that the band annotated as a “10-mer” does not correspond to the decade marker and is very diffuse. This can be attributed to experimental constraints that are challenging to address:

(i) In general there can be discrepancies between the decade marker and RNA migration due to products migrating differently based on their 5' modifications (phosphorylation degree, capping) but also depending on the RNA sequence.

(ii) In addition, we suspect that the composition of the TiLV buffer may interfere with the migration of smaller RNA products, resulting in the artefact at the bottom of the gel. Unfortunately, we have not yet identified which component is causing this effect.

Due to these uncertainties, we have removed the length designation of the “10-mer” and now refer to it as a “Stalled product”. We also provide a schematic below each gel to illustrate the point at to depict the theoretical product of each reaction.

Have these data been replicated and are they reproducible?

Yes, see answer provided to point 5 below.

3. In panel 6C the two bands observed around the 40nt marker are described as products formed by the polymerase either reading through the 5' hook or not. Have the authors considered this may be a single RNA product that was incompletely denatured (despite boiling/urea treatment)? Perhaps the annotation of this product should be changed without additional confirmation of the identity of the product.

In responses to the reviewer's question, we have conducted new assays with a mutated template (v40-S) (Supplementary Fig. 8c, d) and solved a new structure (number 13) at 3.1 Å resolution of TiLV polymerase in a late elongation/pre-termination state stalled with GpCp (Supplementary Fig. 9d,e, Supplementary information 5).

Continuous cryoEM density from the 5' vRNA end to the 3' vRNA end (which binds back to the secondary site) is visible (Supplementary Fig. 9d, e). Only a few nts from the template have to be copied before TiLV could get stalled in a "pre-hook" incorporation state, or proceed up to the 5' end. The mechanism that dictates formation of one or the other product remains unclear.

Both the v40-S assays presented in the new "Supplementary Fig. 8c lane 1-7" are similar to those obtain using the unmutated v40 template in the "Supplementary Fig. 8a lane 4; 8b lanes 6-10".

The structure thus supports the idea of two products, one "pre-hook" and one "post-hook" incorporation.

However, as the reviewer mentioned, due to the uncertainties surrounding the length and identity of the products on the gel, we have revised the annotation to "FL transcription-like products (pre-hook?)/(post-hook?)".

4. Panel 6D, lanes 1 and 2, the text state there are additional stalling sites 2 or 4 nt longer than the expected product. The ladder suggests these products are 1 and 2 nt longer than expected. This discrepancy between the data and text should be addressed.

We agree with the reviewer regarding the discrepancy between the products in the Supplementary Fig. 8a lane 1-2 versus lane 3-6. To simplify the data interpretation, those products have been annotated as due to "read-through / misincorporation", and it is now clearly stated in the text.

5. Lane 7 is suspiciously empty, could this be a misloading? Are the stronger products observed in lanes 9 and 10 reproducible? There is no statement on how reproducible or repeatable this experiment is. By the authors' own admission these in vitro activity data are preliminary.

Those data have been replicated more than three times and are always essentially the same. Indeed, TiLV polymerase elongation structures were successfully obtained based on these *in vitro* assays.

As an example here is another version of the gels now in Supplementary Figure 8a,b.

Concerning, lane 7 (now in Supplementary Fig. 8b), which contains the uncapped primer ending in “...CCA” with all NTPs, we consistently observe no products (see also Supplementary Fig. 8b lane 4, for the same result with a different template). Based on the discussion above regarding the potential inhibition of polymerase initiation by CTP, we hypothesise that the unfavorable primer ending in “...CCA” cannot properly hybridize with the accessible 3' vRNA end “GG” due to the A extension and thus probably does not manage to compete with CTP to bind to the active site, thus decreasing product synthesis when CTP is present in the reaction.

The stronger products observed in lanes 9 and 10 (capped 12- and 13-mer primer), compared to 8 (capped 11-mer) in Supplementary Fig. 8b are not significant (e.g. not reproduced in the gel above), and not obvious in the new assay with a different template (Supplementary Fig. 8d, lane 6 and 7 compared with lane 5). On the other hand, capped 11-, 12- and 13-mer primers consistently give more product than the uncapped 9-mer primer (see Supplementary Figs. 8b,c,d). It is now stated in the text:

“While we do not detect any cap-binding in vitro, reactions with the capped primers show stronger product formation as compared to the 9-mer uncapped primer, which could indicate the optimal hypothetical primer length used by TiLV polymerase for transcription initiation.”

6. *It may be worth removing these data or moving to supplementary/Supplementary. Ideally these data should be replicated.*

As mentioned above, these data have been reproduced several times.

We have also added new assays (Supplementary Figure 8c,d) with a modified template that provide very complementary results, where products mostly conform to theoretical expectations. These data are very important to indicate that the recombinant TiLV polymerase is active for RNA synthesis. Furthermore, they were critical to enable early and late elongation structures to be determined using the 13-mer capped primer. However, as suggested by the author the *in vitro* assays have now been relocated from the main figures to Supplementary Figure 8 (although we think they merit to be in the main figures).

Overall, we have now been very careful to stress the caveats in interpreting the *in vitro* assay data and the whole section now entitled ‘Biochemical demonstration of RNA synthesis activity of recombinant TiLV polymerase’ has been carefully rewritten to take into account the referee’s very relevant remarks. In particular, this paragraph has been added:

‘These preliminary results clearly demonstrate that recombinant TiLV polymerase has primed RNA synthesis activity, also confirmed by the early and late elongation state cryoEM structures described below. However, more detailed analysis is required to identify precisely all the reaction products formed and understand for instance the requirements for efficient de novo initiation in the absence of a priming loop and the mechanism of termination..’

7. *In the cap binding assay promoter RNA was not included. Given that RNA can affect the conformation of the polymerase is it worthwhile to repeat the assay with the addition of vRNA?*

In fact, as described in the Methods, the vRNA promoter was used in the assays for TiLV and influenza A/H7N9. To help the reader, this has now been added in the caption for Fig. 4d.

8. *Similarly, would mutation of PB2/R217A relieve the block to the putative cap binding site and enable cap binding*

To try to address the role of PB2/R217, we cloned and expressed the alanine mutant in insect cells, as suggested by the referee. Unfortunately, TiLV polymerase PB2/R217A was no longer soluble and could not be purified as compared to TiLV polymerase wild-type (see purification gels below).

Figure 1. SDS-gels from TiLV polymerase purification (wild type and PB2 R217A mutant).

This suggests that PB2/R217 is required for correct folding of the PB2 subunit, maybe because of the hydrophobic (stacking) and electrostatic interactions that it is observed to make.

9. *The influenza virus polymerase is known to interact with host proteins (Pol II-CTD and ANP32). Can the authors speculate at all on any possible conservation of the binding sites for interaction with these proteins?*

These are interesting questions that are beyond the scope of the present manuscript and are the subject of our ongoing research.

Minor points

10. *Which influenza virus is being used for comparison to generate residue numbers and amino acid percentages? Generally, the flupol heterotrimer is ~255 kDa not 270 kDa.*

We used the polymerase from bat A/H17N10 (PDB: 4WSB). It is now indicated in the figure captions. The molecular weight has been corrected as well as the comparative percentages and additional typos regarding residue numbers.

11. *How were the Zn sites identified? If it's based on the metal coordination geometry this may be worth stating in the text.*

Indeed, it was by the tetrameric co-ordination with cysteine or histidine residues. This has been added to the text.

12. *Studies on the influenza polymerase from this group and others have shown dimeric polymerase structures, were any of these observed within any of the datasets?*

In the datasets for this paper, no dimers have been observed.

13. *Panel 2D, maybe rather than annotating at "+", the positive control lane could be annotated as FluPol.*

It has been modified accordingly.

14. *Figure 3, there is a line around some of the thumb cyan text that looks different to the others.*

We are not sure where is the line (few "esthetic" modifications have been added to this figure that may have alleviated the problem).

15. *Figure 4, Can the authors specify the PDB of the FluPol being used?*

See also the reply to the reviewer #2:

For the transcriptase conformation, we used PDB:4WSB, which is the polymerase from bat A/H17N10. For the replicase conformation, we previously used PDB:5EPI (polymerase from B/Memphis), now changed to the polymerase from A/H5N1 (PDB:6QPF). This is now indicated in the captions of Fig. 1d and 1e respectively. In Fig. 2, 3, 4a, e, f, we used the polymerase from bat A/H17N10 (PDB: 4WSB). In Fig. 4c, we used the polymerase from A/H7N9 (PDB: 7QTL).

All PDBs numbers have been added in figure legends where needed.

REVIEWERS' COMMENTS

Reviewer #2 (Remarks to the Author):

The authors have very addressed all the concerns raised by me. Their responses seemed thoughtfully considered. Congratulations on a beautiful structure and story!

Reviewer #3 (Remarks to the Author):

I thank the authors for the comprehensive response to my comments and I positively endorse the manuscript.